

# On the numerical integration of the Lorenz-96 model, with scalar additive noise, for benchmark twin experiments

Grudzien Colin[1][2], Bocquet Marc[3], and Carrassi Alberto[2][4]

[1]University of Nevada, Reno, Reno, Nevada, USA
[2]Nansen Environmental and Remote Sensing Center, Bergen, Norway
[4]Mathematical Institute, University of Utrecht, the Netherlands
[3]CEREA, joint laboratory École des Ponts Paris Tech and EDF R&D, Université Paris-Est, Champs-sur-Marne, France

**Correspondence:** Colin Grudzien (cgrudzien@unr.edu)

**Abstract.** Relatively little attention has been given to the impact of discretization error on twin experiments in the stochastic form of the Lorenz-96 equations when the dynamics are fully resolved but random. We study a simple form of the stochastically forced Lorenz-96 equations that is amenable to higher order time-discretization schemes in order to investigate these effects. We provide numerical benchmarks for the overall discretization error, in the strong and weak sense, for several commonly used

integration schemes and compare these methods for biases introduced into ensemble-based statistics and filtering performance. Focus is given to the distinction between strong and weak convergence of the numerical schemes, highlighting which of the two concepts is relevant based on the problem at hand. Using the above analysis, we suggest a mathematically consistent framework for the treatment of these discretization errors in ensemble forecasting and data assimilation twin experiments for unbiased and computationally efficient benchmark studies. Pursuant to this, we provide a novel derivation of the order 2.0

strong Taylor scheme for numerically generating the truth-twin in the stochastically perturbed Lorenz-96 equations.

## 1   Introduction

### 1.1   Twin experiments with geophysical models

Data assimilation and ensemble-based forecasting have together become the prevailing modes of prediction and uncertainty quantification in geophysical modeling. Data assimilation (DA) broadly refers to techniques used to combine numerical model simulations and real-world observations in order to produce an estimate of a posterior probability density for the modeled

state, or some statistic of it. In this Bayesian framework, an ensemble-based forecast represents a sampling procedure for the forecast-prior probability density. The process of sequentially and recursively estimating the distribution for the system's state by combining model forecasts and streaming observations is known as filtering. Due to the large dimensionality and complexity of operational geophysical models, an accurate representation of the true Bayesian posterior is infeasible. Therefore, DA cycles

typically estimate the first two moments of the posterior, or its mode — see, e.g, the recent review of DA by Carrassi et al. (2018).



Many simplifying assumptions are used to produce these posterior estimates and "toy" models are commonly used to assess the accuracy and robustness of a DA scheme's approximations in a controlled environment. Toy models are small-scale analogues to full-scale geophysical dynamics which are transparent in their design and computationally simple to resolve. In this setting it is possible to run rigorous twin experiments in which artificial observations are generated from a "true" trajectory of the toy model, while ensemble-based forecasts are generated and re-calibrated by the DA scheme's observation-analysis-forecast cycle. Using the known "true" system state, techniques for state estimation and uncertainty quantification can be assessed objectively under a variety of model and observational configurations. In the case that: (i) a toy model is entirely deterministic; (ii) both the truth-twin and model-twin are evolved with respect to identical system parameters; (iii) and both the truth-twin and model-twin are resolved with the same discretization; the only uncertainty in a twin experiment lies in the initialization of the model and the observations of the true state. The model dynamics which generate the ensemble forecast are effectively a "perfect" representation of the "true" dynamics which generate the observations (Leutbecher and Palmer, 2008).

The development of toy models and twin experiments has greatly influenced the theory of DA and predictability (Ghil, 2018), and the above perfect-deterministic model configuration has largely driven early results. Lorenz's seminal paper showed that a small loss in the numerical precision of the discretization of the governing equations is sufficient to produce a loss of long term predictability in deterministic chaos (Lorenz, 1963). Understanding that perturbations to a trajectory, tantamount to numerical noise, could lead to rapid divergence significantly influenced the introduction of ensemble-based forecasting in operational settings (Lewis, 2005). In the perfect-deterministic model setting, the asymptotic filter performance can likewise be understood principally in terms of the model's dynamical properties. Particularly, the statistics are determined by the filter's ability to correct for the dynamical instabilities of perturbations along the truth-twin trajectory with respect to the filter's sensitivity to its observations (Gurumoorthy et al., 2017; Bocquet et al., 2017; Bocquet and Carrassi, 2017; Frank and Zhuk, 2018; Maclean and Van Vleck, 2019; Tranninger et al., 2019).

However, the theory for DA and predictability is increasingly concerned with model errors, as studied in, e.g., the recent works of Kang and Harlim (2012); Mitchell and Gottwald (2012); Gottwald and Harlim (2013); Berry and Harlim (2014); Raanes et al. (2015); Carrassi and Vannitsem (2016); Raanes et al. (2018). Model deficiencies in terms of physics which are not fully understood or which are poorly represented prove to be difficult to quantify with an ensemble-based forecast, or to correct with a standard DA cycle. Indeed, when the model is fundamentally biased, increasing the spatial resolution or numerical precision may not generally improve the accuracy of an ensemble-based forecast. It has recently been shown in a deterministic, biased-model setting that the numerical precision of the discretization of the ensemble forecast can be significantly reduced without a major deterioration of the DA cycle's (relative) predictive performance. In this setting, the model bias overwhelms the errors that are introduced due to precision loss when the model twin is resolved with a low order of accuracy; it may be preferable, thus, to exchange lower precision numerics for an increased number of samples in the ensemble-based forecast to better capture the overall spread, see the work of Hatfield et al. (2018).

On the other hand, many aspects of geophysical model uncertainty and variability become tractable in a random and non-autonomous dynamical systems framework, in which certain deficiencies of deterministic models can be mitigated with stochastic forcing (Ghil et al., 2008; Chekroun et al., 2011; Dijkstra, 2013; Ghil, 2017; Boers et al., 2017). In this way, the



theory of random dynamical systems has offered a natural step forward from the perfect-deterministic framework in toy models to the development of novel theory for predictability and DA. As in the perfect-deterministic setting, the DA cycle has recently been given a dynamics-based interpretation in random models to develop new DA methodology (Grudzien et al., 2018a, b).

However, unlike the case of the deterministic, biased-model above, important differences in the statistical properties of model forecasts of stochastic dynamical systems have been observed due to the discretization errors of certain low-order schemes. For example, Frank and Gottwald (2018) develop an order 2.0 Taylor scheme to correct the bias in the drift term induced by the Euler-Maruyama scheme in their study system.

This work similarly studies the effects of the bias on ensemble-based forecasts and the DA cycle due to time-discretization
error in twin experiments in the stochastically perturbed Lorenz-96 system. In the following, we perform an inter-comparison of several commonly used discretization schemes, studying the path-based convergence properties as well as the convergence in distribution of ensemble-based forecasts. The former (strong convergence) determines the ability of the integration scheme to produce observations of the truth-twin consistently with the governing equations; the later (weak convergence) describes the accuracy of the empirically derived sample statistics of the ensemble-based forecast, approximating the fully resolved evolution
of the prior under the Fokker-Plank equations. Using these two criteria, we propose a standard benchmark configuration for the numerical integration of the Lorenz-96 model, with additive noise, for ensemble-based forecasting and DA twin experiments. In doing so, we provide a means to control the bias in benchmark studies intended for environments that have inherent stochasticity in the dynamics, but do not fundamentally misrepresent the physical process. This scenario corresponds to, e.g., an ideal, stochastically reduced model for a multiscale dynamical system, as is discussed in the following.

## 1.2 Stochastic dynamics from multiscale systems

It is a typical (and classical) simplification in filtering literature to represent model error in terms of stochastic forcing in the form of additive or multiplicative noise (Jazwinski, 1970). For many realistic geophysical models, this is actually a reasonable representation of model-reduction errors. Empirically, errors due to coarse grained simulation in probabilistic forecasting are often ameliorated by stochastic parameterization (Palmer et al., 2005). For example, there is evidence that sub-gridscale
convection in oceanic and atmospheric models cannot be accurately parameterized deterministically in terms of the macro-observables of the system. Deterministic parameterizations can faithfully represent the mean response, but typically fail to capture its fast-scale variability, and thus lead to an overall misrepresentation of the large-scale variability of the climate system, see e.g., Gottwald et al. (2016) and references therein.

Stochastic parameterizations thus offer a physically intuitive approach to rectify these issues: many realistic physical pro-
cesses can be considered as noise perturbed realizations of classical deterministic approximations from which they are modeled. Theoretically, unresolved scales can furthermore be reduced to additive Gaussian noise in the asymptotic limit of scale separation due to the Central Limit Theorem (Gottwald et al., 2015). Several mathematically rigorous frameworks have been developed to model and simulate the stochastic forcings to the large-scale dynamics, including averaging methods, perturbation methods and combinations of the two — see, e.g., the survey of approaches by Demaeyer and Vannitsem (2018). In this way,
it is possible to derive exact reductions of deterministic multiscale systems into coarsely resolved, stochastic models.





Mathematically rigorous reductions generally provide an implicit form for the reduced model equations as a mix of deterministic terms, stochastic noise terms and non-Markovian memory terms, all present in the reduced model's equations. This exact, implicit reduction is derived in, e.g., Mori-Zwanzig formalism. In the asymptotic separation of scales, this formulation reduces to a mean field ODE system with additive noise, eliminating the memory terms, describing the system consistently with

homogenization theory. Additionally, empirically based techniques, such as auto-regressive methods, have successfully parameterized model reduction errors (Wilks, 2005; Crommelin and Vanden-Eijnden, 2008). At the state-of-the-art, novel learning techniques are furthermore being developed to construct empirically derived stochastic models that are consistent with mathematical theory for stochastic model reduction, with the goal preserving the underlying model physics (Chorin and Lu, 2015; Vissio and Lucarini, 2018; Cotter et al., 2019).

In this work, we will make a simplifying assumption for the form of the stochasticity: we take the classical filtering framework in which noise is additive, Gaussian, white-in-time and distributed according to a known, scalar covariance matrix. Within the SDE literature, this is sometimes referred to as scalar additive noise, which is a term we will use hereafter. Both the truth-twin and the model-twin will be evolved with respect to the same form for the governing equations, but with respect to (almost-surely) different noise realizations. Conceptually, this represents a perfect-random model; this corresponds physically

to an idealized model for the asymptotic separation of time-scales between the fast and slow layers in the two layer Lorenz-96 model.

### 1.3 The single layer Lorenz-96 model with scalar additive noise

The Lorenz-96 model (Lorenz, 1996) is commonly used in DA literature as a toy model for twin experiments see, e.g., Carrassi et al. (2018)[and references therein]. This is particularly due to the fact that: (i) it is extremely scalable, with the potential to

exhibit spatially extended chaos in high dimensions (Herrera et al., 2011); (ii) it mimics fundamental features of geophysical fluid dynamics, including conservative convection, external forcing and linear dissipation (Lorenz and Emanuel, 1998); and (iii) it can be used in its two-layer form to describe multi-scale dynamics, with a layer of fast variables corresponding to atmospheric dynamics, coupled to a slow layer corresponding to oceanic dynamics (Lorenz, 2005). The two layer form of the Lorenz-96 equations has been of particular interest for developing stochastic parametrizations of sub-grid scale dynamics, see,

e.g., Wilks (2005); Arnold et al. (2013); Chorin and Lu (2015); Lu et al. (2017); Vissio and Lucarini (2018). Likewise, the stochastic reduction of the two layer model to a one layer model has been used to demonstrate techniques for adaptive DA designs in the presence of model uncertainty (Pulido et al., 2018).

In the present study, we consider the single layer form of the Lorenz-96 equations perturbed by additive noise, for which the matrix of diffusion coefficients is a scalar matrix. The classical form for the Lorenz-96 equations (Lorenz, 1996) are defined as

$\frac{\mathrm{d}\boldsymbol{x}}{\mathrm{d}t} \triangleq \boldsymbol{f}(\boldsymbol{x})$, where for each state component $i \in \{1, \cdots, n\}$,

$$f^i(\boldsymbol{x}) = -x^{i-2}x^{i-1} + x^{i-1}x^{i+1} - x^i + F \tag{1}$$

such that the components of the vector $\boldsymbol{x}$ are given by the variables $x^i$ with periodic boundary conditions, $x^0 = x^n$, $x^{-1} = x^{n-1}$ and $x^{n+1} = x^1$. The term $F$ in the Lorenz-96 system, Eq. (1), is the forcing parameter that injects energy to the model. With





the above definition for the classical Lorenz-96 equations, we define the toy model in our consideration to be

$$\frac{\mathrm{d}\boldsymbol{x}}{\mathrm{d}t} \triangleq \boldsymbol{f}(\boldsymbol{x}) + s(t)\mathbf{I}_n \boldsymbol{W}(t),\tag{2}$$

where $\boldsymbol{f}$ is defined as in Eq. (1), $\mathbf{I}_n$ is the $n \times n$ identity matrix, $\boldsymbol{W}(t)$ is an $n$-dimensional Wiener process and $s(t): \mathbb{R} \to \mathbb{R}$ is a measurable function of (possibly) time-varying diffusion coefficients. In the remainder of this work, the system in Eq. (2) will be denoted the **L96-s model**. In contrast to studies in which the objective is to obtain a suitable parameterization of the fast-variable layer of the Lorenz-96 model and perform a model reduction, we use the L96-s system as a "perfect" model
of the known, but random dynamical system of interest. The L96-s model is one particularly simple form for the stochastic Lorenz-96 equations that: (i) expresses the essential randomness; (ii) is a commonly used formulation for filter benchmarks in twin experiments; and (iii) remains amenable to higher order integration schemes for stochastic differential equations.

In appendix A we provide a novel derivation of the strong order 2.0 Taylor method for additive noise (Kloeden and Platen, 2013)[page 359] in the context of the L96-s model. This is a nontrivial derivation of the explicit discretization rule which
has not previously appeared in the literature to the authors' knowledge. We furthermore evaluate the computational efficiency, and rates of convergence for each: (i) Euler-Maruyama / Milstein methods; (ii) the strong order 1.0 Runge-Kutta method; and (iii) the strong order 2.0 Taylor rule. In section 2 we briefly outline each of the different discretization schemes and discuss their modes of convergence. In section 3 we provide our numerical benchmarks of each scheme for convergence and for bias introduced into ensemble-based forecasts and the DA cycle. In section 3.4 we provide a discussion of each of the methods,
and propose a computationally efficient framework for statistically robust twin experiments. Section 4 concludes with a final discussion of results and open questions for future work.

## 2   Numerical simulation of stochastic dynamics for twin experiments

### 2.1   Modes of convergence for stochastic differential equations

Unlike with deterministic models, even when the initial condition of a stochastic dynamical system is precisely known, the
evolution of the state must inherently be understood in a probabilistic sense. This precise initial information represents a Dirac-delta distribution for the prior that is instantaneously spread out due to the unknown realizations of the "true" noise process. In particular, the solution to the initial value problem is not represented by a single sample path, but rather by a distribution derived by the forward evolution with respect to the Fokker-Plank equation. Due to the high numerical complexity of resolving the Fokker-Plank equation in systems with state dimensions greater than 3 (Pichler et al., 2013), a Monte-Carlo, ensemble-based
approach is an appealing alternative and is used to derive empirical statistics of the forward distribution.

However, when the noise realizations are themselves known (as is the case for twin experiments) the path of the delta distribution representing the true system state can be approximately reconstructed using a discretization of the appropriate stochastic calculus. This sample path solution explicitly depends on a particular random outcome and, thereby, we will be interested in criteria for evaluating the discretization error for SDEs that take into account the random variation. For the
numerical integration of the L96-s model, we will consider two standard descriptions of the convergence of solutions to the





approximate, discretized evolution to the continuous-time exact solution — we adapt the definitions from pages $61 - 62$ of Iacus (2009). In each of the below definitions, we refer to the standard Euclidean norm.

**Definition 1.** *Let $\boldsymbol{x}_{\mathrm{SP}}(t)$ be a sample path of an SDE and $\boldsymbol{x}(t)$ be an approximation of $\boldsymbol{x}_{\mathrm{SP}}(t)$ based upon a discretization with a maximum time-step of $\Delta$. Suppose there exists a $\Delta_0 > 0$ such that for any fixed time horizon $T$ and any $0 < \Delta < \Delta_0$,*

$$\mathbb{E}\left[\|\boldsymbol{x}(T) - \boldsymbol{x}_{\mathrm{SP}}(T)\|\right] \leq C\Delta^{\gamma} \tag{3}$$

*where $\mathbb{E}$ denotes the expectation over all possible realizations of the stochastic process, and $C$ is a constant independent of $\Delta$. Then $\boldsymbol{x}(t)$ is said to converge strongly to $\boldsymbol{x}_{\mathrm{SP}}(t)$ on the order $\gamma > 0$.*

We note that in the above definition, $\boldsymbol{x}$ and $\boldsymbol{x}_{\mathrm{SP}}$ are both subject to the same outcomes of the random process, and the expectation is taken over all possible outcomes.

Strong convergence is the analogue of the discretization of a non-stochastic trajectory and is used to judge the accuracy (on average) of reconstructing a specific sample path based upon a known realization of a Brownian motion. We may also consider, however, whether a discretization rule is able to accurately represent a statistic of the forward evolved distribution when estimated over many sample paths with respect to different realizations of Brownian a motion — this is the motivation for weak convergence.

**Definition 2.** *Let $\boldsymbol{x}_{\mathrm{SP}}(t)$ be a sample path of an SDE and $\boldsymbol{x}(t)$ be an approximation of $\boldsymbol{x}_{\mathrm{SP}}(t)$ based upon a discretization with a maximum time-step of $\Delta$. Suppose there exists a $\Delta_0 > 0$ such that for any fixed time horizon $T$, any $2(\gamma + 1)$ continuously differentiable function $g$ of at most polynomial growth and any $0 < \Delta < \Delta_0$,*

$$\|\mathbb{E}\left[g(\boldsymbol{x}(T)) - g(\boldsymbol{x}_{\mathrm{SP}}(T))\right]\| \leq C\Delta^{\gamma}, \tag{4}$$

*where $\mathbb{E}$ denotes the expectation over all possible realizations of the stochastic process, and $C$ is a constant independent of $\Delta$. Then $\boldsymbol{x}(t)$ is said to converge weakly to $\boldsymbol{x}_{\mathrm{SP}}(t)$ on the order $\gamma > 0$.*

The distinction between strong and weak convergence can be thought of as: (i) strong convergence measures the mean of the path-discretization errors over all sample paths; whereas (ii) weak convergence can measure the error when representing the mean of all sample paths from an empirical distribution. When studying the empirical statistics of a stochastic dynamical system or of an ensemble-based forecast, weak convergence is an appropriate criterion for the discretization error. However, when we study the RMSE of a filter in a twin experiment, we assume that we have realizations of an observation process depending on a specific sample path of the governing equations. Therefore, while the accuracy of the ensemble-based forecast may be benchmarked with weak convergence, strong convergence is the appropriate criterion to determine the consistency of the truth-twin with the governing equations (Hansen and Penland, 2006).

We will now introduce several commonly studied methods of simulation of stochastic dynamics and discuss their strengths and their weaknesses. To limit the scope of the current work, we will focus only on strong discretization schemes; while a strong discretization scheme will converge in both a strong and weak sense, weak discretizations do not always guarantee convergence





in the strong sense. We note here, however, that it may be of interest to study weak discretization schemes solely for the purpose of the efficient generation of ensemble-based forecasts – this may be the subject of a future work, and will be discussed further in section 4. We will introduce the discretization rules in a general form whenever appropriate. In this section, however, we will only discuss the strong order 2.0 Taylor scheme in a reduced form derived specifically for the Lorenz-96 system with scalar additive noise (L96-s). A more general formulation for additive noise, and its reduction to our model, is the content of appendix A.

## 2.2 A general form of stochastic differential equations

Consider a generic stochastic differential equation **(SDE)** of the form

$$\mathrm{d}\boldsymbol{x} = \boldsymbol{f}(\boldsymbol{x},t)\mathrm{d}t + \mathbf{S}(\boldsymbol{x},t)\,\mathrm{d}\boldsymbol{W}(t) \tag{5}$$

where $\boldsymbol{f}$ is a vector valued map in $\mathcal{C}^2(\mathbb{R}^n \times \mathbb{R}, \mathbb{R}^n)$, $\boldsymbol{W}(t)$ is an $n$-dimensional Wiener process, and $\mathbf{S}$ is a matrix valued map of diffusion coefficients in $\mathcal{C}^2(\mathbb{R}^n \times \mathbb{R}, \mathbb{R}^{n \times n})$, equal to a square root of the covariance function of the Gaussian stochastic process $\mathbf{S}(\boldsymbol{x},t)\boldsymbol{W}(t)$. In general, the diffusion coefficients $\mathbf{S}(\boldsymbol{x},t)$ can depend on both the state of the random process and time. We note, however, that in the case of additive noise $\big($when $\mathbf{S}(\boldsymbol{x},t) \equiv \mathbf{S}(t)\big)$ the derivative of the diffusion coefficients with respect to $\boldsymbol{x}$ is zero and the Itô and Stratonovich drift coefficients are equal, due to the zero adjustment term (Kloeden and Platen, 2013)[see page 109]. In the following discussions we will denote $\boldsymbol{x}_k \triangleq \boldsymbol{x}(t_k)$ and assume that uniform time-steps are taken such that $t_{k+1} \triangleq t_k + \Delta$.

## 2.3 Euler-Maruyama and Milstein schemes

The Euler-Maruyama scheme is among the simplest extensions of deterministic integration rules to discretize systems of SDEs such as Eq. (5). Like the standard deterministic order 1.0 Euler scheme, the Euler-Maruyama scheme approximates the evolution of a sample path by a functional relationship expressed by Eq. (5)

**Euler-Maruyama:**

$$\boldsymbol{x}_{k+1} = \underbrace{\boldsymbol{x}_k + \boldsymbol{f}(\boldsymbol{x}_k,t_k)\Delta}_{a} + \underbrace{\mathbf{S}(\boldsymbol{x}_k,t_k)\boldsymbol{W}_\Delta}_{b}, \tag{6}$$

where the term (6.a) is the deterministic Euler scheme and the term (6.b) is the matrix of diffusion coefficients $\mathbf{S}$ multiplied by the noise $\boldsymbol{W}_\Delta$, a mean zero, Gaussian random vector of covariance $\Delta\mathbf{I}_n$.

The Euler-Maruyama scheme benefits from its simple functional form, adaptability to different types of noise and its intuitive representation of the SDE. However, with the Definitions 1 and 2 in mind, it is important to note that Euler-Maruyama generally has a weak order of convergence of 1.0, but a strong order of convergence only of 0.5 (discussed by, e.g., Kloeden and Platen (2013) in Theorem 10.2.2). The loss of one half order of convergence from the deterministic Euler scheme arises from the differences of deterministic calculus and Itô calculus.

The Milstein scheme includes a correction to the rule in Eq. (6), adjusting the discretization of the stochastic terms to match the first order Itô-Taylor expansion. In the case that the matrix of diffusion coefficients $\mathbf{S}(\boldsymbol{x},t)$ is diagonal, let us denote the





$i$-th diagonal element as $S^i(\boldsymbol{x}, t)$; then the Milstein scheme takes the component-wise form,

**Milstein scheme for diagonal noise:**

$$x^i_{k+1} = x^i_k + f^i(\boldsymbol{x}_k, t_k)\Delta + S^i(x_k, t_k)W^i_\Delta \tag{7a}$$

$$+ \frac{1}{2}S^i(\boldsymbol{x}_k, t_k)\frac{\partial S^i}{\partial x^i}\left[\left(W^i_\Delta\right)^2 - \Delta\right] \tag{7b}$$

for the $i$-th state component, where the partial derivative is evaluated at $(\boldsymbol{x}_k, t_k)$. This and the general multi-dimensional form for the Milstein scheme can be found in pages 345 - 348 of Kloeden and Platen (2013). We note that, in the case of additive noise, the partial derivatives $\frac{\partial S^i}{\partial x^i}$ vanish and the Euler-Maruyama and Milstein scheme are equivalent; thus in our example the Euler-Maruyama scheme achieves a strong order of convergence at $1.0$.

Although the Euler-Maruyama scheme is simple to implement, we shall see in the following that the cost of achieving mathematically consistent simulations quickly becomes prohibitive. Despite the fact that it achieves both a strong and weak convergence on the order 1.0 in the L96-s model, the overall discretization error is significantly higher than even other order 1.0 strong convergence methods — the difference between the Euler-Maruyama scheme and other methods lies in the constant $C$ in the bounds in definitions 1 and 2.

### 2.4 Runge-Kutta Methods

The convergence issues for the Euler-Maruyama scheme are well understood and there are many rigorous methods to overcome its limitations (Kloeden and Platen, 2013)[see pages xxiii - xxxvi and references therein]. However, general higher order methods can become quite complex due to: (i) the presence of higher order Itô-Taylor expansions in Itô forms of SDEs; and/or (ii) the approximation of multiple Itô or Stratonovich integrals necessary to resolve higher order schemes. Stochastic Runge-Kutta methods can at least eliminate the higher order Itô-Taylor expansions, though do not automatically deal with the issues around multiple stochastic integrals; for a discussion on the limits of higher order Runge-Kutta schemes see, e.g., Burrage and Burrage (1996, 1998).

Given a system of SDEs as in Eq. (5), the straightforward extension of the classical four-stage Runge-Kutta method, proven by Rüemelin (1982), is given as





**Strong order 1.0 Runge-Kutta**

$$\boldsymbol{\kappa}_1 \triangleq \boldsymbol{f}\left(\boldsymbol{x}_k, t_k\right)\Delta + \mathbf{S}\left(\boldsymbol{x}_k, t_k\right)\boldsymbol{W}_\Delta \tag{8a}$$

$$\boldsymbol{\kappa}_2 \triangleq \boldsymbol{f}\left(\boldsymbol{x}_k + \frac{\boldsymbol{\kappa}_1}{2}, t_k + \frac{\Delta}{2}\right)\Delta$$
$$+ \mathbf{S}\left(\boldsymbol{x}_k + \frac{\boldsymbol{\kappa}_1}{2}, t_k + \frac{\Delta}{2}\right)\boldsymbol{W}_\Delta \tag{8b}$$

$$\boldsymbol{\kappa}_3 \triangleq \boldsymbol{f}\left(\boldsymbol{x}_k + \frac{\boldsymbol{\kappa}_2}{2}, t_k + \frac{\Delta}{2}\right)\Delta$$
$$+ \mathbf{S}\left(\boldsymbol{x}_k + \frac{\boldsymbol{\kappa}_2}{2}, t_k + \frac{\Delta}{2}\right)\boldsymbol{W}_\Delta \tag{8c}$$

$$\boldsymbol{\kappa}_4 \triangleq \boldsymbol{f}\left(\boldsymbol{x}_k + \boldsymbol{\kappa}_3, t_k + \Delta\right)\Delta$$
$$+ \mathbf{S}\left(\boldsymbol{x}_k + \boldsymbol{\kappa}_3, t_{k+1} + \Delta\right)\boldsymbol{W}_\Delta \tag{8d}$$

$$\boldsymbol{x}_{k+1} \triangleq \boldsymbol{x}_k + \frac{1}{6}\left(\boldsymbol{\kappa}_1 + 2\boldsymbol{\kappa}_2 + 2\boldsymbol{\kappa}_3 + \boldsymbol{\kappa}_4\right). \tag{8e}$$

The straightforward extension of the four-stage Runge-Kutta scheme to stochastic systems in Eq. (8) has the benefits that: (i) it is an intuitive extension of the commonly used four-stage deterministic rule, making the implementation simple; (ii) it makes few assumptions on the structure of the governing equations; (iii) and in the small noise limit, the rule will be compatible with the deterministic order 4.0 implementation. However, in trade-off with generality, by not exploiting the dynamical system structure, the discretization in Eq. (8) has strong order of convergence 1.0 (Hansen and Penland, 2006). Alternatively, we may consider using, e.g., the strong order 2.0 Runge-Kutta method for scalar additive noise (Kloeden and Platen, 2013)[page 411]; the form of the L96-s equations indeed satisfies this condition. However, this is an implicit method, coming with the additional cost of, e.g., Newton-Raphson iterations to solve each step forward. While this is a necessary measure for stiff equations, the L96-s equations do not demand this type of precision.

## 2.5 Strong order 2.0 Taylor scheme

As a generic "out-of-the-box" method for numerical simulation, the four-stage Runge-Kutta method in Eq. (8) has many advantages over Euler-Maruyama, and is a good choice when the noise is non-additive or the deterministic part of the system lacks structure that leads to simplification. On the other hand, the combination of: (i) constant and vanishing second derivatives of the Lorenz-96 model; (ii) the rotational symmetry of the system in its spatial index; and (iii) the condition of scalar additive noise; together allow us to present the strong order 2.0 Taylor rule as follows.

**Strong order 2.0 Taylor:**

Define the constants $\rho$ and $\alpha$ as

$$\rho = \frac{1}{12} - \frac{1}{2\pi^2}; \qquad\qquad \alpha = \frac{\pi^2}{180} - \frac{1}{2\pi^2}. \tag{9}$$

For each step of size $\Delta$:





1. Draw 5 random vectors, $\boldsymbol{\xi}, \boldsymbol{\zeta}, \boldsymbol{\eta}, \boldsymbol{\phi}, \boldsymbol{\mu} \sim N(\mathbf{0}, \mathbf{I}_n)$, independently and identically distributed (iid).

2. Compute the random vectors

$$\boldsymbol{a} = -2\sqrt{\Delta}\rho\boldsymbol{\mu} - \frac{\sqrt{2\Delta}}{\pi}\boldsymbol{\zeta}, \tag{10a}$$

$$\boldsymbol{b} = \sqrt{\Delta}\alpha\boldsymbol{\phi} + \sqrt{\frac{\Delta}{2\pi^2}}\boldsymbol{\eta}. \tag{10b}$$

3. For $\boldsymbol{\xi}$, $\boldsymbol{a}$ and $\boldsymbol{b}$ defined as above, and for each entry of the random vectors indexed by $l, j$, define the coefficients

$$\begin{aligned}\Psi^p_{(l,j)} \triangleq &\frac{\Delta^2}{3}\xi_l\xi_j + \frac{\Delta^{\frac{3}{2}}}{4}(\xi_l a_j + \xi_j a_l) + \frac{\Delta}{2}a_l a_j \\ &- \frac{\Delta^{\frac{3}{2}}}{2\pi}(\xi_l b_j + \xi_j b_l),\end{aligned} \tag{11}$$

which are used to define the random vectors,

$$\boldsymbol{\Psi}^p_+ \triangleq \begin{pmatrix} \Psi^p_{(n,2)} \\ \vdots \\ \Psi^p_{(n-1,1)} \end{pmatrix}, \qquad\qquad \boldsymbol{\Psi}^p_- \triangleq \begin{pmatrix} \Psi^p_{(n-1,n)} \\ \vdots \\ \Psi^p_{(n-2,n-1)} \end{pmatrix}, \tag{12}$$

and

$$\boldsymbol{J}^p_\Delta \triangleq \begin{pmatrix} \frac{\Delta}{2}\left(\sqrt{\Delta}\xi_1 + a_1\right) \\ \vdots \\ \frac{\Delta}{2}\left(\sqrt{\Delta}\xi_n + a_n\right) \end{pmatrix}. \tag{13}$$

Then, in matrix form, the integration rule from time $t_k$ to $t_{k+1}$ is given as

$$\boldsymbol{x}_{k+1} = \boldsymbol{x}_k + \boldsymbol{f}\Delta + \frac{\Delta^2}{2}\nabla\boldsymbol{f} \cdot \boldsymbol{f} \tag{14a}$$

$$+ s_k\sqrt{\Delta}\boldsymbol{\xi} + s_k\nabla\boldsymbol{f} \cdot \boldsymbol{J}^p_\Delta + s_k^2\left(\boldsymbol{\Psi}^p_+ - \boldsymbol{\Psi}^p_-\right). \tag{14b}$$

where the term (14a) is the deterministic order 2.0 Taylor scheme, the term (14b) is the stochastic component of the rule and $s_k \triangleq s(t_k)$. Deriving this relatively simple form for the strong order 2.0 Taylor rule for the L96-s system is non-trivial and this is explained in detail in appendix A.

In the following section we will illustrate several numerical benchmarks of each of the methods, describing explicitly their rates of strong and weak convergence. We note here that the order 2.0 Taylor scheme above takes a very different practical form in the following where we perform these benchmarks, versus the case of a twin experiment. If simulating a sample path for a twin experiment, one can simply use the functional relationships described above to *define* the linear combinations of Karhunen-Loève Fourier coefficients for the Brownian bridge process between the discretization points, see Eq. (A33). Particularly, the simulation of a sample path by the above converges to *some* sample path, consistent with the governing equations.





On the other hand, when we numerically benchmark the Taylor scheme discretization of a *known sample path*, the depen-
dence of the scheme on the Fourier coefficients for the Brownian bridge in between discretization times excludes the possibility
of the direct implementation above. Particularly, we must use the *known* realization of the Brownian motion rather than simu-
lating the Brownian bridge in between discretization times. The random vector $\boldsymbol{b}$ above is defined functionally in Eq. (A9) as
an infinite sum of vectors of random Fourier coefficients, such that there is a dependence on the order of truncation $p$ of the
term $\boldsymbol{b}$ in our benchmarks. This again differs substantially from the case in which we wish to simulate some *arbitrary* path with
the Taylor scheme; in this case, the relationship between $\boldsymbol{b}$ and the Fourier coefficients is defined by an analytical, functional
relationship (so this term is exact) while it is sufficient to make a truncation order $p = 1$ to other terms to maintain order 2.0
strong convergence. For general purposes in twin experiments and ensemble forecasting, the form in Eq. (14) is the one of
interest and is simple to implement. We will detail the experimental difference for the numerical discretization benchmarks in
the following section.

## 3 Numerical benchmarks

### 3.1 Benchmarks for strong and weak convergence

We begin with benchmarks of each method's strong and weak convergence. Subsequently, we will evaluate the differences
in the ensemble-based forecast statistics of each method, as well as biases introduced in DA twin experiments. Using these
numerical benchmarks, we will formulate a computationally efficient framework for controlling the discretization errors in
twin experiments.

In each of the strong and weak convergence benchmarks we use the same experimental setup. As a matter of computational
convenience, we set the system dimension at $n = 10$. This allows us to study the asymptotic filtering statistics, both in number
of samples in ensemble-based forecasts as well as in the temporal limit and the number of analyses. We begin by generating
a long, climatological trajectory of the L96-s model, using the order 2.0 Taylor scheme with a time-step of $\Delta = 10^{-3}$. This
solution is spun onto the climatological statistics, using $5 \times 10^6$ integration steps, and subsequently evenly sampled at an
interval of $\delta_t = 2$, generating $M = 500$ unique initial conditions. We denote this collection of initial conditions $\{\boldsymbol{v}_m(s)\}_{m=1}^M$.
The parameter $s$ is a fixed diffusion coefficient for Eq. (2), fixed for each experiment with $s \in \{0.1, 0.25, 0.5, 0.75, 1.0\}$.

For each initial condition $\boldsymbol{v}_m(s)$ above, we generate a unique batch of $N = 10^2$ realizations of a 10-dimensional Brownian
motion process; each Brownian motion realization, indexed by $b = 1, \cdots, N$, is defined by a matrix $\mathbf{W}_m^b(s) \in \mathbf{R}^{n \times 10^6}$. Each
component of the matrix $\mathbf{W}_m^b(s)$ is drawn iid from $N\left(0, 10^{-7}\right)$ representing the realized Brownian motion over the interval
$[0, 0.1]$, discretized at an interval of $\Delta_{\mathrm{SP}} = 10^{-7}$. With respect to this discretized Brownian motion realization, we generate a
reference trajectory which we denote as $\mathbf{x}_{\mathrm{SP}}$; this sample path is generated with the Euler-Maruyama scheme as a simple and
transparent approximation of the "true" path corresponding to the realized Brownian motion $\mathbf{W}_m^b(s)$, converging to the true
sample path strongly at an order of $10^{-7}$.

For each initial condition $\boldsymbol{v}_m(s)$, and each unique choice of $\mathbf{W}_m^b(s)$ as above, we subsequently generate coarsely discretized
approximations of the reference trajectory $\boldsymbol{x}_{\mathrm{SP}}$, using each of the: (i) Euler-Maruyama; (ii) Runge-Kutta; and (iii) Taylor





schemes as in section 2, however, only with a discretization of size $\{\Delta_q = 10^{-q}\}_{q=1}^{3}$. Each coarse approximation utilizes the identical realization of the Brownian motion used to generate the reference path, but discretized according to the time-step $\Delta_q$. To obtain the appropriate Brownian motion realization at the coarse time-steps, we take the component-wise sum of all finely

discretized Brownian increments in between the coarse steps to define the realized motion discretized at these points.

Note that the L96-s model is spatially homogeneous and the Euclidean norm of its state depends on the dimension of the system $n$. Therefore, we estimate the strong and weak convergence of the discretization schemes componentwise, independently of the state dimension of the model. Let $x^i$ and $x^j$ denote the $i$-th and $j$-th component of the vector $\boldsymbol{x} \in \mathbb{R}^n$ respectively. For each initial condition $\boldsymbol{v}_m(s)$, we approximate the following expected values for an arbitrary component $j$:

$$\mathbb{E}\left[\left|x^j(T) - x^j_{\mathrm{SP}}(T)\right|\right] \approx \frac{1}{N}\sum_{b=1}^{N}\sqrt{\sum_{i=1}^{n}\frac{\left[x^i(T) - x^i_{\mathrm{SP}}(T)\right]^2}{n}}, \tag{15}$$

$$\left|\mathbb{E}\left[x^j(T) - x^j_{\mathrm{SP}}(T)\right]\right| \approx \sqrt{\sum_{i=1}^{n}\frac{\left[\frac{1}{N}\sum_{b=1}^{N}x^i(T) - x^i_{\mathrm{SP}}(T)\right]^2}{n}}, \tag{16}$$

where $T = 0.1$ It is to be understood in the above, despite the suppressed indices, that the difference of the reference solution $\boldsymbol{x}_{\mathrm{SP}}$ and the coarsely discretized solution $\boldsymbol{x}$ depend on the same value $s$ for the diffusion, the same initial condition $m$ and the same realized Brownian motion $b$. Thus the above equations make an approximation of: Eq. (15) – strong convergence with the

batch average root mean square error (RMSE) of the coarsely discretized solution versus the reference; and Eq. (16) – weak convergence with the RMSE of the batch mean of the coarsely discretized solutions versus the batch mean of the reference solutions.

It is known that the batch average error on the right-hand-side of Eqs. (15 - 16) is Gaussian distributed around the true expected value, on the left-hand-side, for batches of Brownian motions with $N > 15$; computing these batch average errors

over the $M = 500$ different initial conditions, we can compute the sample-based estimate of the expected value (the mean over all batches) and the standard deviation of the batch average to compute confidence intervals for the expectation (Kloeden and Platen, 2013)[see section 9.3].

For each coarse discretization, with step size $\{\Delta_q = 10^{-q}\}_{q=1}^{3}$, we compute the point estimate for each expected value on the left-hand-side of Eqs. (15-16) with the average of the right-hand-side of Eqs. (15-16) taken over the $M = 500$ initial

conditions. The average of the right-hand-side of of Eqs. (15-16) over all batches will be denoted as the point estimate $\epsilon_{\Delta_q}$. Then, within log-log scale, base-10, we compute the best-fit line between the points $\left\{\left(\Delta_q, \epsilon_{\Delta_q}\right)\right\}_{q=1}^{3}$ using weighted-least-squares, with weights proportionate to the inverse of the batch-average, sample standard deviation of the Eqs. (15 - 16). The slope of the line estimated as above is our approximation of the order of convergence $\gamma$ and the intercept is the constant $C$.

Given a known realization of a Brownian motion, generating the finely discretized sample path between the coarse discretiza-

tion times $\Delta_q$, the direct implementation of the Taylor rule in Eq. (14) is no longer appropriate. In particular, this rule assumes that there is an unknown Brownian Bridge process (Kloeden and Platen, 2013)[see pages 42 - 43] between the discretization times for which we can generate the random Fourier coefficients independently of the Brownian motion at discretization points. As a modification of the Taylor scheme, utilizing the *known* realization of the Brownian Bridge process between the discretiza-





tion steps $\Delta_q$, we compute the Fourier coefficients of the Brownian Bridge, up to $p$-th order, directly by the right-Reimann

sums as in Eq. (A6 - A7). Because the term $b_j$ in Eq. (A9) is defined by an infinite sum of the random Fourier coefficients, computed in Eq. (A7), this modification necessarily leads to an approximation not present in the rule in Eq. (14). In particular, the rule in Eq. (14) utilizes the analytical, functional relationships in Eq. (A33) to compute the term $b_j$ for *some unknown* Brownian Bridge. However, with respect to all following benchmarks, we found no significant difference in performance when directly computing only the order $p = 1$ and the order $p = 25$ Fourier coefficients as described above, so we present only the

$p = 1$ case for simplicity.

In Fig. 1, we plot the point estimates for the discretization error, measured in strong convergence, for each of the discretization methods: (i) Euler-Maruyama; (ii) Runge-Kutta; and (iii) Taylor, as compared with the finely discretized reference solution. It is immediately notable in this plot that while each order of strong convergence is empirically verified, the lines for Runge-Kutta and Taylor schemes actually cross. Indeed, the estimated slope $\gamma$ matches the theoretical value within a $10^{-2}$ decimal

approximation for each scheme ($\gamma = 1.0$ for each Euler-Maruyama and Runge-Kutta, $\gamma = 2.0$ for Taylor), but the constants in the order analysis play a major role in the overall discretization error in this system: the Taylor rule has convergence on order 2.0 in the size of the discretization steps but the constant $C$ associated with the Taylor rule penalizes this scheme much more heavily than in the case of the Runge-Kutta scheme. The constant $C$ for the Runge-Kutta scheme reduces the overall order of discretization error by about an order of magnitude for each diffusion level (see Table 1).

The effect of this constant $C$ is even more prominent for the case of weak convergence, pictured in Fig. 2. In particular, for the weaker diffusion levels of $s = 0.1$ and $0.25$, although the order of weak convergence is 1.0 in the size of discretization step, the order 2.0 Taylor rule fails to achieve a superior discretization error than the Runge-Kutta scheme in this regime. Here, the constant $C$ for the Runge-Kutta scheme is extremely small, lowering the overall order of discretization error by two orders of magnitude, despite the order 1.0 weak convergence (see Table 1).

Most interestingly, as the diffusion level $s$ approaches zero, it appears that the behavior of the Runge-Kutta scheme converges in some sense to the four-stage (order 4.0) deterministic Runge-Kutta scheme, but the difference in their orders is reflected in the constant $C$. Analytically the stochastic Runge-Kutta rule coincides with the deterministic rule in the zero-noise limit. This analytical convergence is also true for the Taylor scheme, where the zero noise limit is the deterministic Taylor scheme, but the order of convergence for both the stochastic and deterministic Taylor schemes are 2.0 and the overall error does not change so

dramatically.

It should be noted that the standard deviation of the batch discretization error of the Runge-Kutta scheme is much higher with a time-step of 0.1 than with smaller steps sizes for both strong and weak convergence metrics. Particularly, this leads to the point estimate for $\Delta_q = 0.1$ receiving much less weight in the line fitting, resulting in the disconnected line and point estimates for the Runge-Kutta scheme. The theoretical order of convergence estimates rely on an assumption that the step size

is taken sufficiently small, and therefore, the weighted fitting of the slope and the intercept for these lines is more accurate in estimating the asymptotic discretization errors.



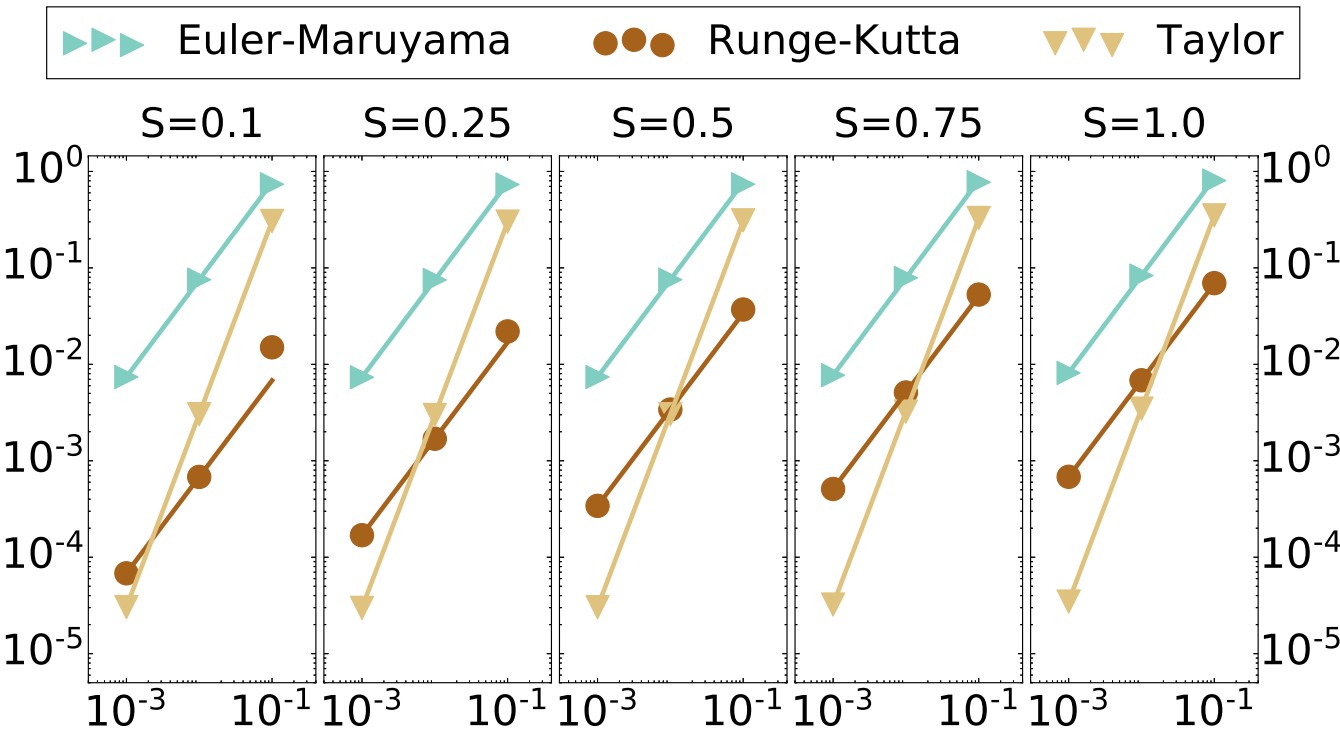

**Figure 1. Strong convergence benchmark**. Vertical axis - discretization error, log scale. Horizontal axis - step size, log scale. Diffusion level $s$. Length of time $T = 0.1$.

**Table 1.** Estimated discretization error constant $C$. The constant corresponds to the expected bound of the discretization error, $C\Delta^{\gamma}$, where $\Delta$ is the maximum time-step of the discretization and is sufficiently small. $\gamma$ equals 1.0 for Euler-Maruyama and Runge-Kutta, and 2.0 for Taylor. Values of $C$ are rounded to $\mathcal{O}\left(10^{-2}\right)$.

| Diffusion | $s = 0.1$ | | $s = 0.25$ | | $s = 0.5$ | | $s = 0.75$ | | $s = 1.0$ | |
|---|---|---|---|---|---|---|---|---|---|---|
| Scheme/ Mode | Strong | Weak | Strong | Weak | Strong | Weak | Strong | Weak | Strong | Weak |
| Euler-Maruyama | 7.85 | 7.85 | 7.81 | 7.80 | 7.86 | 7.82 | 8.22 | 8.14 | 8.66 | 8.53 |
| Runge-Kutta | 0.07 | 0.01 | 0.17 | 0.02 | 0.34 | 0.03 | 0.51 | 0.05 | 0.69 | 0.08 |
| Taylor | 30.50 | 30.50 | 29.84 | 29.79 | 30.52 | 30.34 | 32.38 | 31.98 | 34.81 | 34.15 |





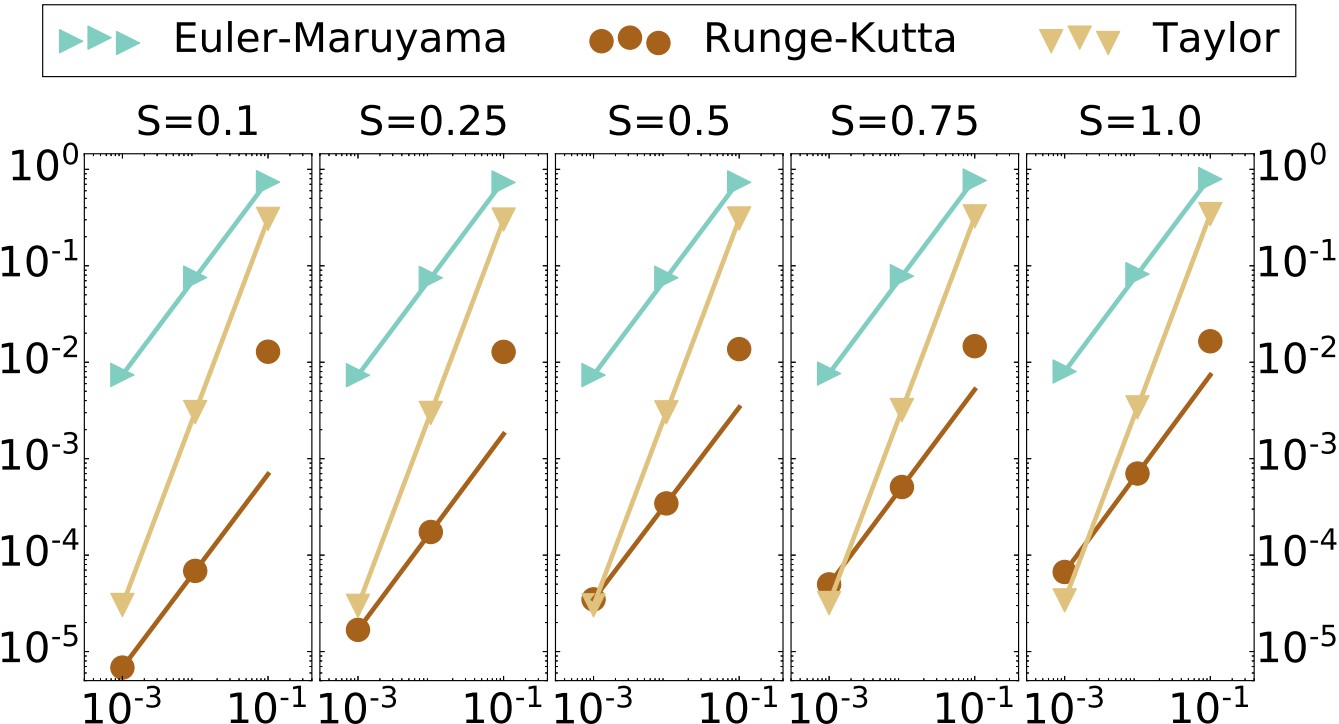

**Figure 2. Weak convergence benchmark**. Vertical axis - discretization error, log scale. Horizontal axis - step size, log scale. Diffusion level $s$. Length of time $T = 0.1$.

## 3.2 Ensemble forecast statistics

Given the consistent performance of the order 2.0 Taylor scheme using a maximal step size of $\Delta = 10^{-3}$, with both strong and weak discretization error on the order of $\mathcal{O}\left(10^{-4}\right)$, we will use this configuration as a benchmark setting to evaluate the other methods. While the Runge-Kutta scheme often has better performance than the Taylor scheme in the overall weak-discretization error, the level of discretization error also varies on one order of magnitude between different diffusion settings. Therefore, in the following experiments we will consider how the different levels of discretization error and diffusion affect the empirically generated, ensemble-based forecast statistics in the L96-s model with respect to a consistent reference point.

We sample once again the initial conditions $\{v_m(s)\}_{m=1}^M$ as described in section 3.1. For each initial condition, we generate a unique batch of $N = 10^2$ realizations of a 10-dimensional Brownian motion process. Once again, the Brownian motion realizations are indexed by $b = 1, \cdots, N$, where each realization is defined by a matrix, $\mathbf{W}_q^b(s) \in \mathbf{R}^{n \times 2*10^5}$. Each component of the matrix $\mathbf{W}_m^b(s)$ is drawn iid from $N\left(0, 10^{-3}\right)$; this represents a Brownian motion realized over the interval $[0, 20]$, discretized at an interval of $\Delta = 10^{-3}$. For $g \in \{e, r, t\}$ let the matrix $\mathbf{X}_g(t) \in \mathbb{R}^{n \times N}$ be defined as the ensemble matrix. Let the vector $\overline{\boldsymbol{x}}_g(t) \triangleq \frac{1}{N} \sum_{i=1}^N \mathbf{X}_g^b(t)$ be defined as the ensemble mean at time $t$, averaged over all realized Brownian motions $\left\{\mathbf{W}_m^b(s)\right\}_{b=1}^N$, where the ensemble is generated by the Euler-Maruyama ($g = e$), Runge-Kutta ($g = r$) or Taylor ($g = t$) scheme respectively.





Then, for an arbitrary ensemble $g \in \{\mathrm{e}, \mathrm{r}, \mathrm{t}\}$, we define the spread at time $t$ to be

$$\mathrm{spread}_g(t) \triangleq \sqrt{\frac{1}{N-1}\sum_{b=1}^{N}\frac{\left(\overline{\boldsymbol{x}}_g(t) - \mathbf{X}_g^b(t)\right)^{\mathrm{T}}\left(\overline{\boldsymbol{x}}_g(t) - \mathbf{X}_g^b(t)\right)}{n}}, \qquad (17)$$

i.e., defined by the unbiased, sample-based estimate of the standard deviation of the mean-square-deviation of the anomalies (Whitaker and Loughe, 1998).

For each the Euler-Maruyama and Runge-Kutta schemes ($g \in \{\mathrm{e}, \mathrm{r}\}$), we measure: (i) the root-mean-square deviation of the ensemble mean from the benchmark

$$\mathrm{RMSD}_g(t) \triangleq \sqrt{\sum_{i=1}^{n}\frac{\left(\overline{\boldsymbol{x}}_g^i(t) - \overline{\boldsymbol{x}}_{\mathrm{t}}^i(t)\right)^2}{n}}; \qquad (18)$$

and (ii) the ratio of the ensemble spread with that of the benchmark $(\mathrm{spread}_g(t)/\mathrm{spread}_{\mathrm{t}}(t))$; each at an interval of $\delta_t = 0.01$ over the forecast $t \in [0, 20]$. The integration step size for the Euler-Maruyama and the Runge-Kutta schemes are varied over

$\{\Delta_q = 10^{-q}\}_{q=2}^{3}$, and the diffusion level is varied $s \in \{0.1, 0.25, 0.5, 0.75, 1.0\}$.

In Figs. 3 - 4, we plot: (i) the median as a solid line; (ii) the inner 80-percentile as a shaded region; and (iii) the minimum and maximum values attained as dashed lines; where each summary statistic is computed over the $M = 500$ initial conditions. In each of the figures we plot the RMSD of the ensemble mean versus the benchmark and the ratio of the ensemble spread only over the interval $[0, 10]$; we find that the statistics are stable in the interval $[10, 20]$ and we neglect the longer time series which

remain approximately at the same values as those shown at the end of the pictured time series.

In Fig. 3 we note immediately that there are no major differences between the ensemble mean or spread of the finely discretized Runge-Kutta scheme and the benchmark Taylor scheme for forecasts up to length $T \in [0, 3]$. Indeed, in all diffusion regimes, the RMSD of the ensemble means is bounded by $0.11$ for forecasts up to length $T = 3$, and below $0.1$ for forecasts up to length $T = 2.8$. In the higher diffusion regimes, in which the weak-discretization error of the Runge-Kutta scheme becomes

slightly higher than the benchmark system, we notice a slight change in the performance in which the ensemble means begin to deviate earlier; however, the difference occurs well beyond what would be considered a practical upper limit of a forecast length, around $T = 2$. Asymptotically as the forecast length $T \to \infty$, the RMSD of the means settles to small variations around the median value of $0.5$, indicating that by $T = 10$ the Runge-Kutta and benchmark Taylor ensembles have become close to their climatological distributions. This is indicated likewise in the ratio of the ensemble spreads where, beyond $T = 10$, the

width of the percentiles around the median ratio, at $1.0$, becomes steady.

The relatively slow divergence of the ensembles under Runge-Kutta and Taylor discretizations, finally reaching similar climatological distributions, sits in contrast to the ensemble statistics of the Euler-Maruyama scheme. Notably, the ensemble mean of the Euler-Maruyama scheme quickly diverges. Moreover, at low-diffusion values, the short-time-scale divergence is also consistently greater than the deviation of the climatological means. This indicates that, unlike with the Runge-Kutta

scheme, a strong bias is present in the empirical forecast statistics with respect to the Euler-Maruyama scheme. The ensemble-based climatological mean generated with the Euler-Maruyama scheme is similar to that under the Taylor scheme; however, the spread of the climatological statistics is consistently greater than that of the benchmark system. After the short period of





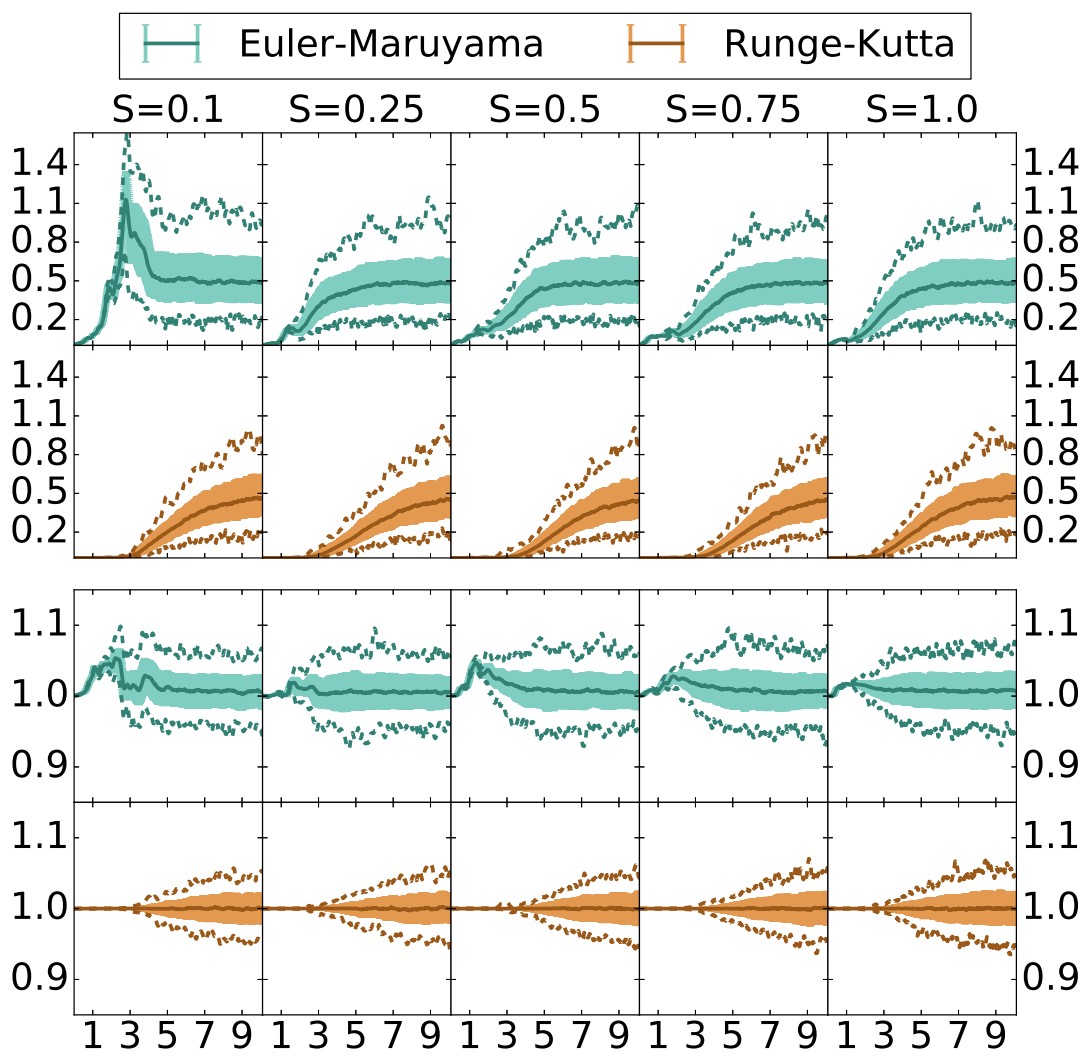

**Figure 3. Ensemble forecast statistics deviation over time — fine discretization.** Euler-Maruyama and Runge-Kutta discretized with time-step $\Delta = 10^{-3}$. **Top:** RMSD. **Bottom:** ratio of spread. Median – solid. Inner $80\%$ – shaded. Min/max – dashed.





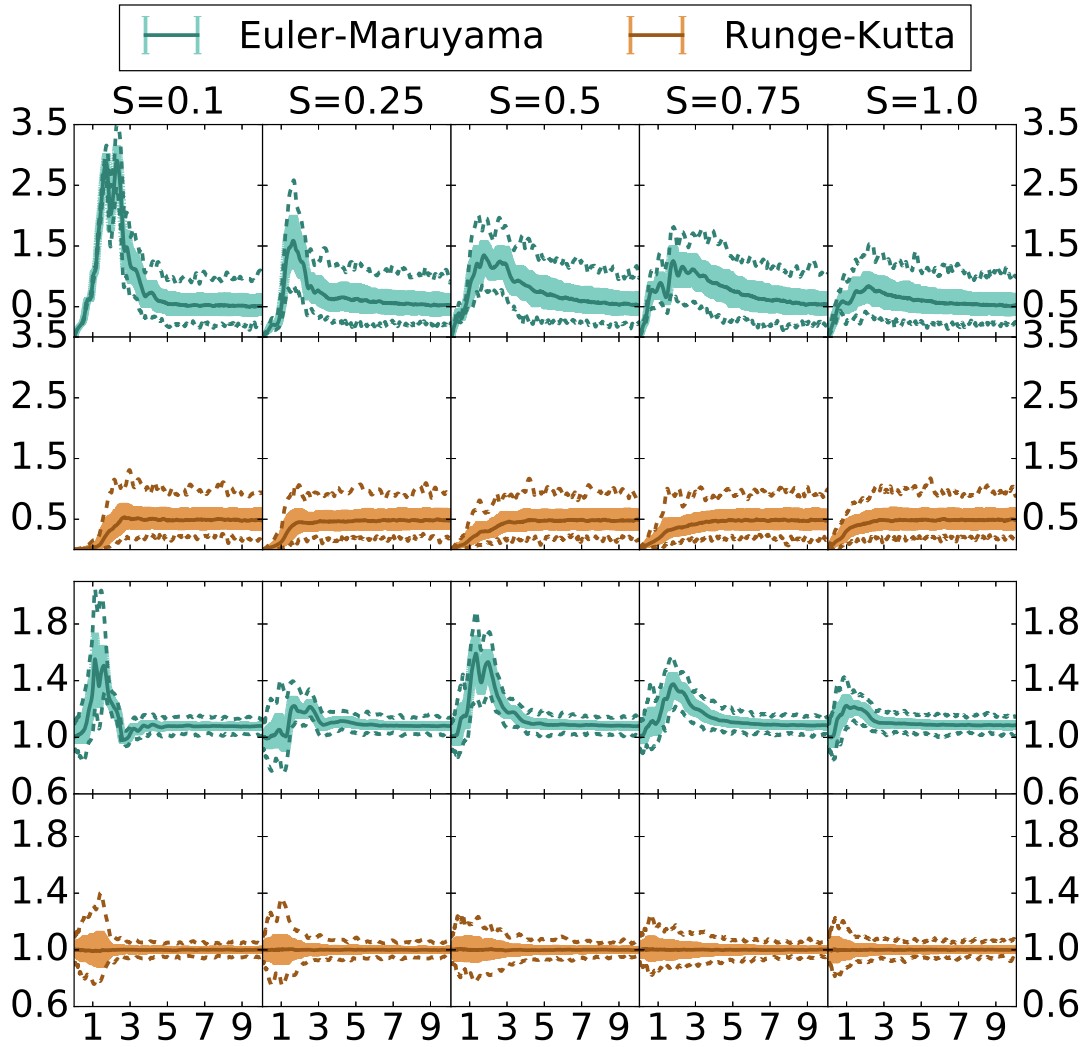

**Figure 4. Ensemble forecast statistics deviation over time — coarse discretization.** Euler-Maruyama and Runge-Kutta discretized with time-step $\Delta = 10^{-2}$. **Top:** RMSD. **Bottom:** ratio of spread. Median – solid. Inner $80\%$ – shaded. Min/max – dashed.

divergence, the median ratio of the spread of the Euler-Maruyama ensemble versus the Taylor ensemble is actually consistently above 1.0.

Increasing the step size of the Euler-Maruyama and Runge-Kutta ensembles to $\Delta = 10^{-2}$, we see in Fig. 4 some similar patterns and some differences. With the large step size the divergence of the ensemble means has a faster onset. However, particularly for the Euler-Maruyama scheme we see the presence of a bias, indicated by the large short-time-scale deviation, substantially greater than the climatological deviation of means. For both the Euler-Maruyama and the Runge-Kutta scheme, increased diffusion shortens the initial period of the divergence of the ensemble means, bringing each ensemble closer to the





climatological statistics more rapidly. With the increased discretization error, the Runge-Kutta scheme has more variability in its ensemble spread, but remains unbiased with the median ratio of spreads centered at one. For the Euler-Maruyama scheme, however, we see the artificial increase in the ensemble spread even more pronounced, with the climatological ratio almost always greater than 1.0, with the minimum value for the ratio of spreads often exceeding this.

Given the above results, we can surmise that the Runge-Kutta scheme will be largely unbiased in producing ensemble-based
forecast statistics, with maximum time discretization of $\Delta \in (0, 0.01]$. At the upper endpoint of this interval, divergence of the means occurs more rapidly and there is more variation in the ensemble spread versus a finer step size. However, the result is to settle more quickly onto the climatological statistics, close to the benchmark Taylor system. The short term and climatological statistics of the Euler-Maruyama scheme, however, suffer from biases especially in low-diffusion regimes or for a maximal time discretization on the order of $\mathcal{O}\left(10^{-2}\right)$. In the following section, we will explore how these observed differences in
ensemble-based statistics of these schemes affects the asymptotic filtering statistics.

### 3.3  Data assimilation twin experiments

Here we study the RMSE and the spread of the analysis ensemble of a simple, stochastic (perturbed observation) ensemble Kalman filter (EnKF) (Evensen, 2003). We fix the number of ensemble members at $N = 100$ and set the L96-s system state to be fully observed (with Gaussian noise) for all experiments such that no additional techniques are necessary to ensure filter
stability for the benchmark system. Particularly, in this configuration, neither inflation nor localization are necessary to ensure stability — this is preferable because inflation and localization techniques typically require some form of tuning of parameters to overcome the usual rank deficiency, the associated spurious correlations and over-confidence in the ensemble-based covariance estimates (Carrassi et al., 2018). The benchmark system uses the order 2.0 Taylor scheme with a time discretization of: (i) the truth-twin with $\Delta_t = 10^{-3}$; and (ii) the ensemble with $\Delta_e = 10^{-3}$.

In Fig. 5, we plot the asymptotic-average EnKF analysis RMSE and ensemble spread for our benchmark system over a range of diffusion levels $s$ and a range of observational error variance $r$, where $s, r \in \{0.1, 0.25, 0.5, 0.75, 1.0\}$. The filter's analysis RMSE is evaluated in terms of the analysis ensemble mean estimate versus the truth-twin, with observations given at length $T = 0.1$ between analyses. The average RMSE and spread are calculated over $2.5 \times 10^4$ analysis cycles, with an initial $5 \times 10^3$ analysis cycles pre-computed, not contributing to the average, as a spin-up for the filter to reach its stable statistics. We see, in
all combinations of the model and observational error parameters, that the filter is performing well when compared with the standard deviation of the observation errors. The spread is also consistently at comparable values to the RMSE, indicating that the performance of the EnKF is stable in this regime (Whitaker and Loughe, 1998).

### 3.3.1  Varying the ensemble integration method for data assimilation

In this section, we will compare several different DA twin experiment configurations with the benchmark system, in which the
Taylor scheme generates the truth-twin and model forecast with a fine time step. The configuration which is compared to the benchmark system will be referred to as the "test" system. We fix the truth-twin to be generated in all cases by the order 2.0 Taylor scheme, with time-step $\Delta_t = 10^{-3}$. We will vary, on the other hand, the method of generating the forecast ensemble



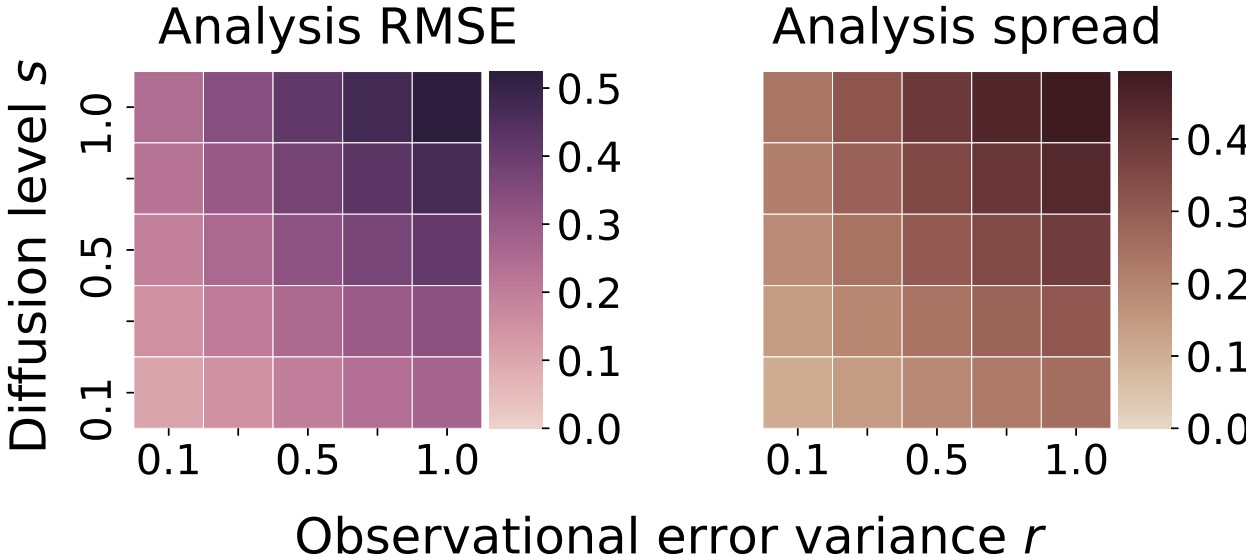

**Figure 5. Truth: Taylor** $\Delta_t = 10^{-3}$**/ Ensemble: Taylor** $\Delta_e = 10^{-3}$**:** asymptotic-average analysis ensemble RMSE and spread of benchmark configuration. Vertical axis – level of diffusion $s \in \{0.1, 0.25, 0.5, 0.75, 1.0\}$. Horizontal axis – variance of observation error $r \in \{0.1, 0.25, 0.5, 0.75, 1.0\}$.

for the test system with different choices of discretization schemes and the associated time-step. For each choice of ensemble integration scheme, we once again compute the asymptotic-average analysis RMSE and spread over $2.5 \times 10^4$ analyses, with

a $5 \times 10^3$ analysis spin-up such as to reach stable statistics.

We drop the phrase "asymptotic-average analysis" in the remaining portions of section 3 and instead refer to these simply as the RMSE and spread. In each of the following figures, we plot (i) the RMSE of the EnKF generated in the test system, minus the RMSE of the benchmark system; and (ii) the ratio of the spread of the EnKF generated with the test system compared with that of the benchmark configuration. All filters are supplied identical Brownian motion realizations for the model errors,

which are used to propagate the ensemble members with their associated integration schemes. Likewise, identical observations (including randomly generated errors) and observation perturbations in the stochastic EnKF analysis are used for each filter at corresponding analysis times.

As was suggested by the results in section 3.2, the difference between the RMSE and the ratio of the spreads for the configuration in which the ensemble is generated by the Runge-Kutta scheme with step size $\Delta_e = 10^{-3}$ and the benchmark

DA configuration is nominal: the RMSE difference is at the order $\mathcal{O}\left(10^{-6}\right)$, with a mean value and standard deviation on the order $\mathcal{O}\left(10^{-7}\right)$ across the configurations; the ratio of the spread differs from 1.0 at an order of $\mathcal{O}\left(10^{-6}\right)$, with a mean value on the order of $\mathcal{O}\left(10^{-6}\right)$ and a standard deviation on the order of $\mathcal{O}\left(10^{-7}\right)$. For these reasons, we do not plot the comparison of the ensemble generated with the Runge-Kutta scheme with step size $\Delta_e = 10^{-3}$ and the benchmark configuration. The





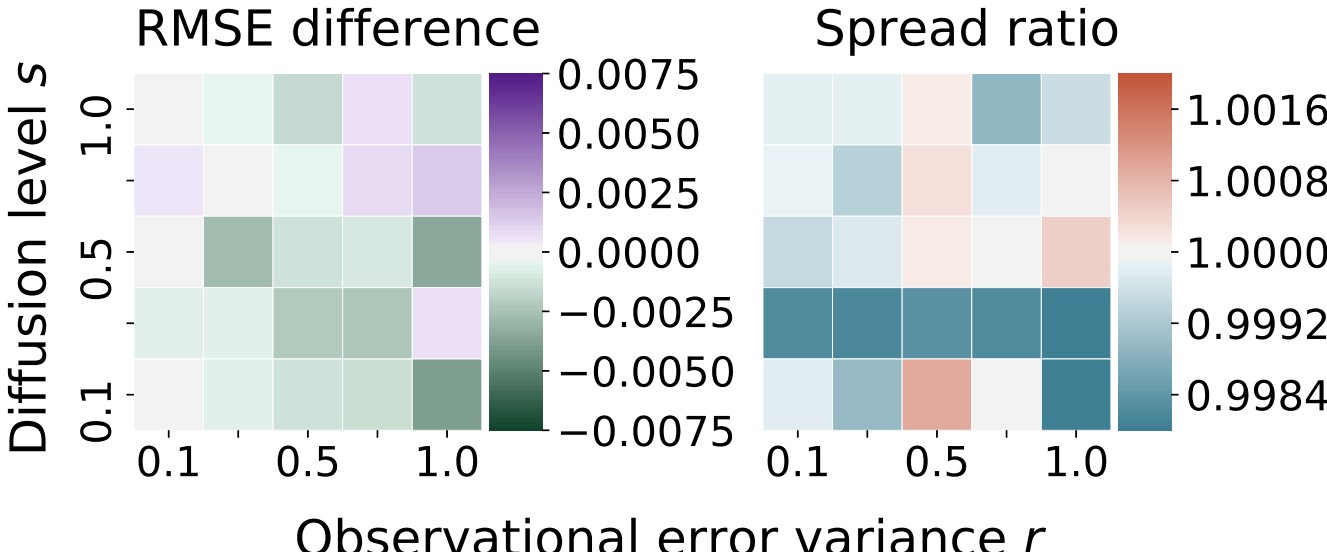

**Figure 6. Truth: Taylor** $\Delta_t = 10^{-3}$ **/ Ensemble: Runge-Kutta** $\Delta_e = 10^{-2}$**:** difference of RMSE / ratio spread with benchmark Fig. 5.

next question is whether increasing the time-step of the Runge-Kutta scheme to $\Delta_e = 10^{-2}$ will impact the filter performance,

especially in terms of causing bias in the filter's results.

     In Fig. 6, the test system uses the Runge-Kutta scheme with coarse time-step $\Delta_e = 10^{-2}$. We find what appears to be small, random variation in the difference, where in some cases the coarse time-step scheme slightly out-performs the benchmark system in terms of the RMSE. These RSME differences, however, appear to be effectively unstructured in sign or magnitude with regards to the diffusion level $s$ and the observational error variance $r$, indicating that this amounts to random numerical

fluctuation and is mostly unbiased; this is likewise the case for the ratio of the spread. The average of these RMSE differences is approximately $-8 \times 10^{-5}$, with a standard deviation of approximately $10^{-3}$; the ratio of spreads differs from 1.0 on average by about $4 \times 10^{-5}$ with a standard deviation of approximately $8 \times 10^{-4}$. With these differences being very slight, we expect that increasing the integration time-step for the Runge-Kutta scheme to $\Delta_e = 10^{-2}$ will not introduce any structural biases to twin experiments based on the discretization error.

We are secondly interested in seeing how the Euler-Maruyama scheme generating the ensemble compares with the benchmark system when using a maximal time-step of $\Delta_e \in \{10^{-2}, 10^{-3}\}$. In Fig. 7, the test system uses the Euler-Maruyama scheme with time-step $\Delta_e = 10^{-3}$. Note, the scale for the RMSE difference in Fig. 7 matches the scale of the positive differences in Fig. 6. However, the scale for the spread ratio in Fig. 7 differs from the scale in Fig. 6 by about an order of magnitude. We find that in contrast to the coarse grained Runge-Kutta scheme, there is indeed structure in this plot, similar to the results in

section 3.2. For low levels of diffusion, there is a clear bias introduced by the Euler-Maruyama scheme in which the ensemble is artificially inflated, and as well has a lower overall accuracy (though by a small measure). However, it is also of interest that the performance of the Euler-Maruyama scheme and the benchmark system are almost indistinguishable for higher levels of





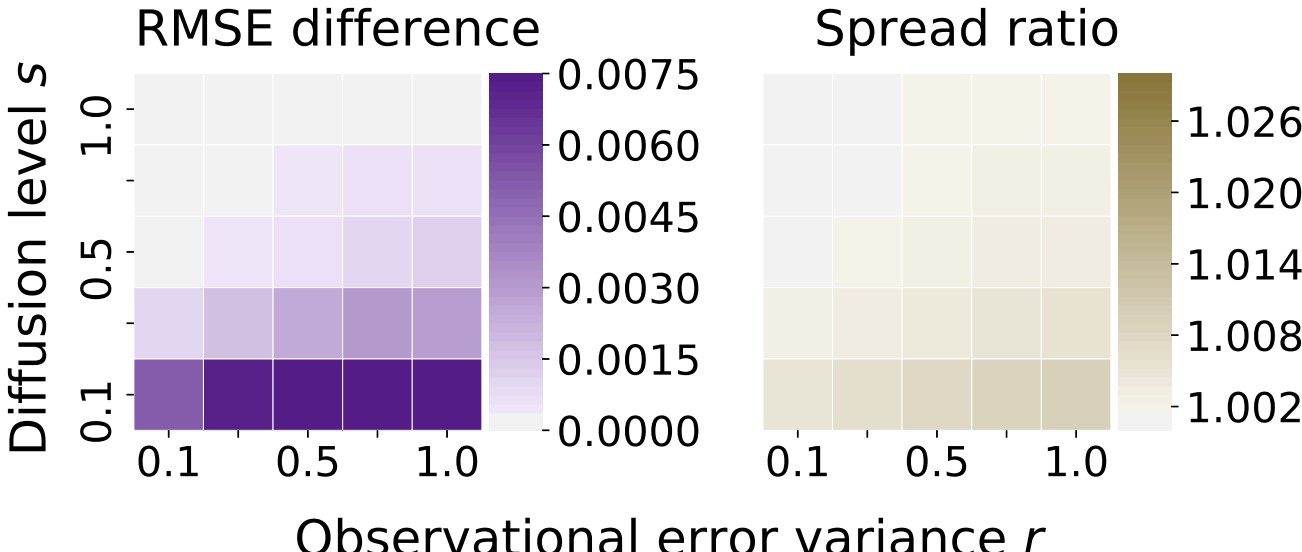

**Figure 7. Truth: Taylor** $\Delta_t = 10^{-3}$**/ Ensemble: Euler-Maruyama** $\Delta_e = 10^{-3}$**:** difference of RMSE / ratio of spread with benchmark Fig. 5.

diffusion. With a time-step of $\Delta = 10^{-3}$, the Euler-Maruyama scheme achieves comparable performance with the benchmark approach for high levels of diffusion; however, there is clearly a bias introduced that systematically affects the filters' accuracy

in a low-diffusion regime, in contrast with the last example.

Next we turn our attention to Fig. 8, where the test system uses the Euler-Maruyama scheme with a time-step of $\Delta_e = 10^{-2}$. Here, a log scale is introduced in the measure of the RMSE difference, and a new linear scale for the spread ratio. We see the same structure of the bias introduced to the filter, where for low-diffusion levels there is a strong bias, sufficient to cause filter divergence in this configuration. However, for high-diffusion levels, this bias is less significant and the filter performance

is roughly comparable to the benchmark system, with the difference being on the order of $10^{-2}$ for $s \geq 0.5$. We again see the artificial effect of the inflation due to the Euler-Maruyama scheme in the spread of the ensemble, with the same structure present as in the last example. In Fig. 8, the scale for the spread is also about an order of magnitude larger than in Fig. 7.

### 3.3.2 Varying the truth-twin accuracy for data assimilation

Finally, we examine the effect of lowering the accuracy of the truth-twin on the test system's filter performance relative to

the benchmark configuration. In each of the following figures, we again compare the RMSE and spread of the benchmark configuration in Fig. 5 — in all cases the test system will generate the truth-twin using the order 2.0 Taylor scheme, with a coarser time step of $\Delta_t = 5 \times 10^{-3}$. In this case, based on the estimate from Table 1, the discretization error for the truth-twin is approximately equal to $8 \times 10^{-4}$.





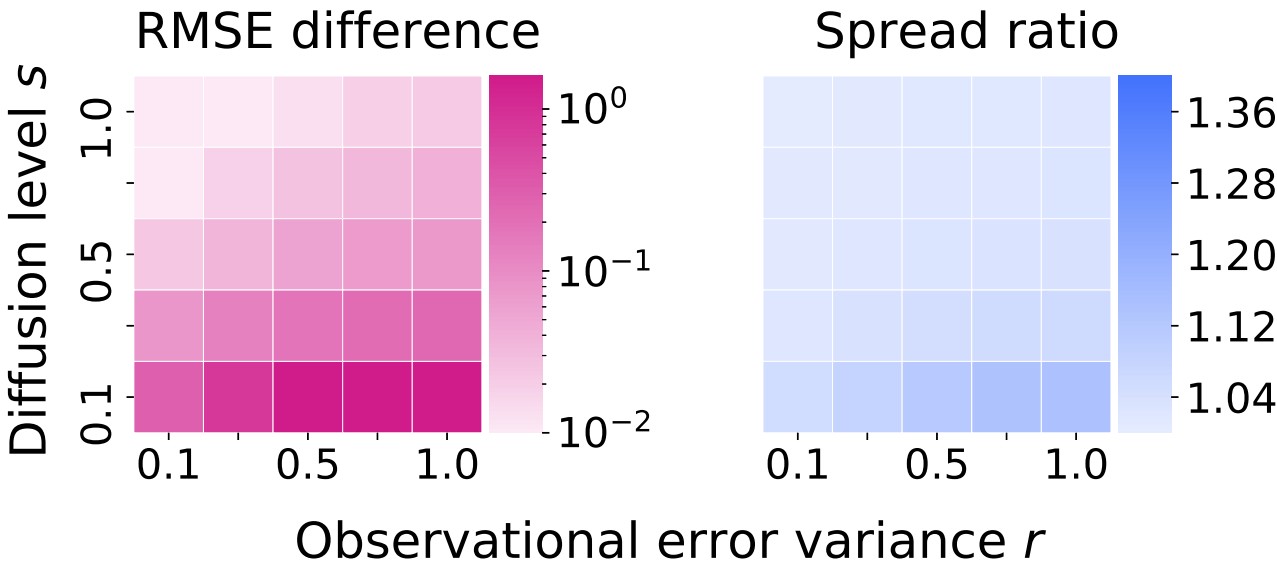

**Figure 8. Truth: Taylor** $\Delta_t = 10^{-3}$**/ Ensemble: Euler-Maruyama** $\Delta_e = 10^{-2}$**:** difference of RMSE / ratio of spread with benchmark Fig. 5.

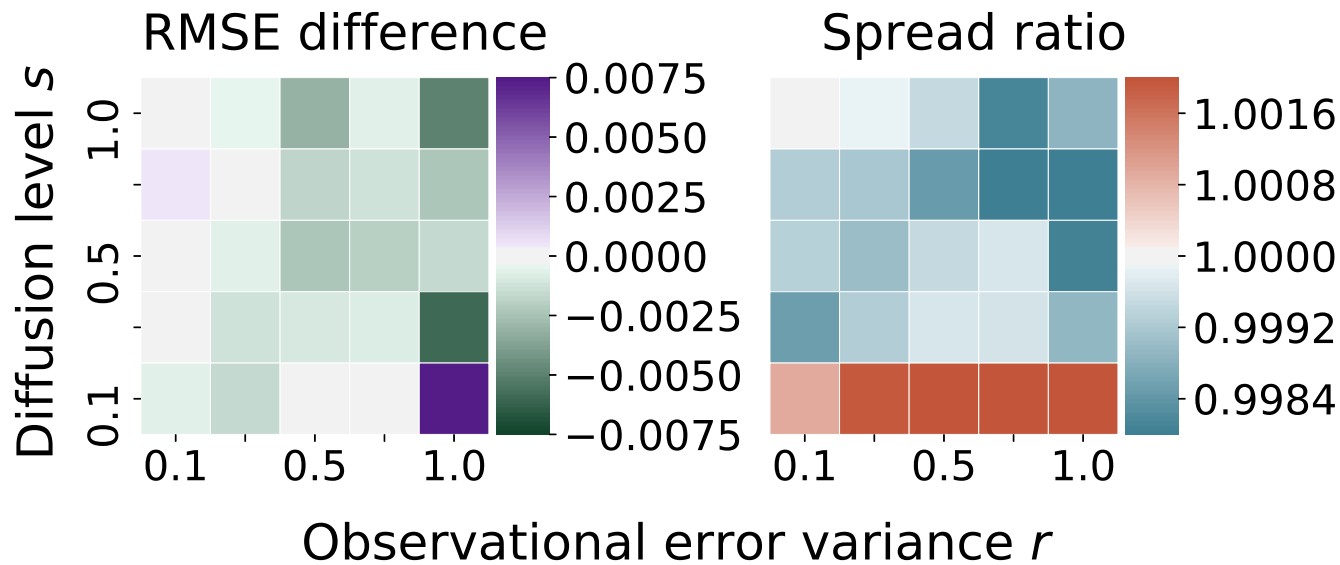

**Figure 9. Truth: Taylor** $\Delta_t = 5 \times 10^{-3}$**/ Ensemble: Runge-Kutta** $\Delta_e = 10^{-2}$**:** difference of RMSE / ratio of spread with benchmark Fig. 5.





In Fig. 9, the Runge-Kutta scheme generates the ensemble with a step size of $\Delta_e = 10^{-2}$. The figure uses identical scales

for both RMSE and spread as in Fig. 6. The filtering statistics, with reduced accuracy of the truth-twin in conjunction with the reduced accuracy Runge-Kutta scheme for the ensemble, are largely the same as in the case of the more accurate truth-twin. The discretization error in the generation of the ensemble forecast is on the order of at most $8 \times 10^{-4}$, but is usually lower for the smaller diffusion levels (see Table 1). The main distinction lies in that there is a clear separation of the spread ratio into the low-diffusion regime in which the spread of the coarse system is higher while the spread of the coarse system is lower for

high-diffusion regime. The mean of the RMSE differences in Fig. 9 is approximately $-5 \times 10^{-5}$, while the standard deviation is approximately $3 \times 10^{-3}$. The difference of the spread from 1.0 is approximately $3 \times 10^{-4}$ on average, with a standard deviation of approximately $2 \times 10^{-3}$.

As a final comparison with the benchmark system, in Fig. 10 the test system generates the ensemble by the Euler-Maruyama scheme with time-step $\Delta_e = 10^{-3}$. The scale for the RMSE differences is the same as in Fig. 6, while the scale for the ratio of

the spread is the same as in Fig. 7. We note that the qualitative structure of the differences is close to that in Fig.7, with a notable difference. Here, the difference from the benchmark system at high-diffusion levels is relaxed, and the EnKF generated by the fine grained evolution under Euler-Maruyama at times performs better than the benchmark system, when there is the additional discretization error of the truth-twin. This may correspond to the fact that the discretization error for the truth-twin under the Taylor scheme is slightly higher with the higher diffusion levels. However, the overall bias introduced by the Euler-Maruyama

scheme into the twin experiment seems to remain largely the same. We neglect a plot comparing the system in which the ensemble is generated by the Euler-Maruyama scheme with step $\Delta_e = 10^{-2}$ — this case is largely the same as results in Fig. 8, with a similar pattern of filter divergence at low diffusion and relaxation at higher diffusion.

### 3.4    An efficient framework for twin experiments

We briefly consider the computational complexity of the Euler-Maruyama scheme in Eq. (6), the strong order 1.0 stochastic

Runge-Kutta scheme in Eq. (8) and the strong order 2.0 Taylor scheme in Eq. (A39). We note that every one of these methods applied in the L96-s system has a per-iteration complexity that grows linearly in the system dimension $n$. This is easy to see for the Euler-Maruyama scheme and is verified by, e.g., Hansen and Penland (2006) for the stochastic Runge-Kutta scheme. On the other hand, it may appear that the numerical complexity of one iteration of the Taylor scheme is $\mathcal{O}(n^2)$ due to the multiplication of the vectors $\boldsymbol{f}$ and $\boldsymbol{J}_\Delta^p$ with the Jacobian $\nabla \boldsymbol{f}$. However, for any $n \geq 4$, there are only four nonzero elements

in each row of $\nabla \boldsymbol{f}$; the sparsity of the Jacobian means that an efficient implementation of the matrix multiplication will only grow in complexity at $\mathcal{O}(n)$.

However, there are significant differences in the number of iterations necessary to maintain a target discretization error over an interval $[0, T]$. A typical forecast length for a twin experiment in the L96-s system of $T \in [0.1, 0.5]$, corresponding to weakly and strongly nonlinear behavior respectively at the endpoints of this interval. The necessary number of iterations to produce

a truth-twin with discretization error on $\mathcal{O}\left(10^{-6}\right)$ as in the usual Lorenz-96 system is around $\mathcal{O}\left(10^2\right)$ to $\mathcal{O}\left(10^3\right)$ integration steps with the strong order 2.0 Taylor scheme. This is because, even with the order 2.0 strong convergence, the Taylor scheme





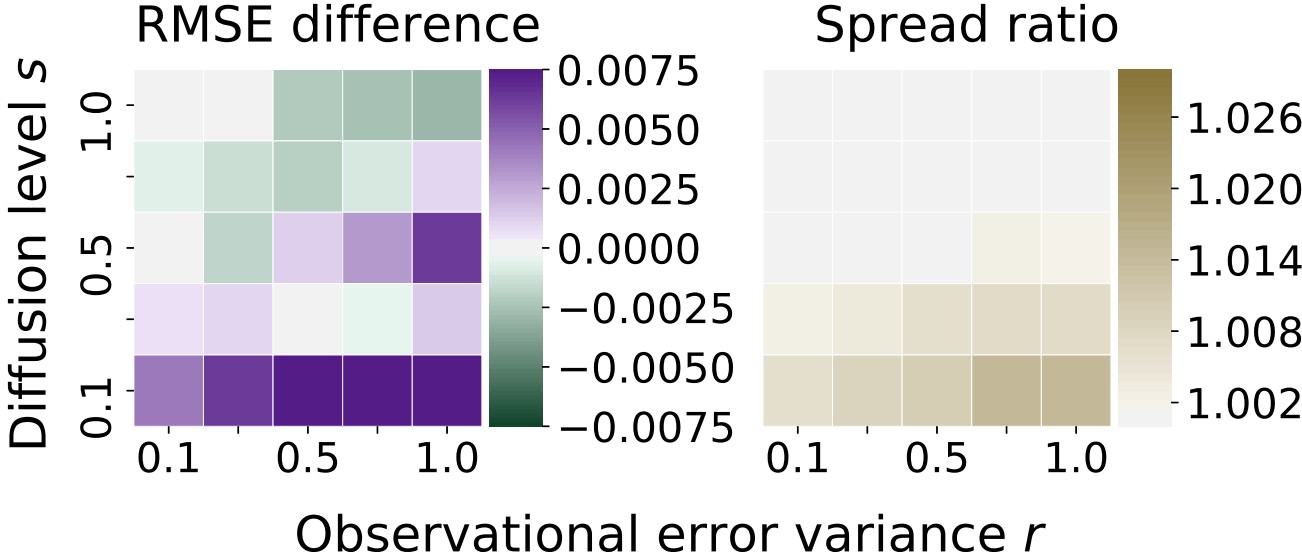

**Figure 10. Truth: Taylor** $\Delta_t = 5 \times 10^{-3}$ **/ Ensemble: Euler-Maruyama** $\Delta_e = 10^{-3}$**:** difference of RMSE / ratio of spread with benchmark Fig. 5.

has to compensate for the large constant term by dropping to a maximal step size of $\mathcal{O}\left(10^{-4}\right)$. As a practical compromise, we suggest a higher target discretization error bounded by $10^{-3}$.

The order 2.0 Taylor scheme, with a maximal step size of $\Delta_t = 5 \times 10^{-3}$, achieves a strong discretization error bounded by

$10^{-3}$ across all diffusion regimes. This level of strong discretization error is not possible with either the Euler-Maruyama or Runge-Kutta schemes without dropping the maximal step size to at most $10^{-3}$, making the order 2.0 Taylor scheme a suitable choice for generating the truth-twin. On the other hand, in ensemble-based DA, the greatest numerical cost in a twin experiment lies in the generation of the ensemble forecast. Across the diffusion regimes, from weak $s = 0.1$ to strong $s = 1.0$, we have seen that the stochastic Runge-Kutta scheme achieves a weak discretization error bounded by $10^{-3}$ when the maximal step

size is $\Delta_e = 10^{-2}$. This suggests the use of a hybrid approach to simulation in which the Taylor and Runge-Kutta schemes are used simultaneously for different scopes.

The combination of: (i) Truth-twin – Taylor, $\Delta_t = 5 \times 10^{-3}$; (ii) model-twin – Runge-Kutta $\Delta_e = 10^{-2}$ maintains the target discretization error bounded by $10^{-3}$ with relatively few computations. Moreover, using the Runge-Kutta scheme to generate the ensemble has the benefit that it is easy to formulate in vectorized code over the ensemble. In sections 3.2 and 3.3, we

demonstrated that this configuration does not fundamentally bias the ensemble forecast or the DA cycle as compared with more accurate numerical discretizations. While there were noted small differences in some of the short term forecast statistics, the climatological statistics remain largely the same. Likewise, the difference in the asymptotic filtering statistics appear to be tantamount to numerical noise when compared with a more accurate configuration. Given the results in section 3.2 on





the differences for the short-range ensemble forecast statistics with the coarsely resolved Runge-Kutta scheme, we expect the
conclusions to hold for standard DA twin experiments with forecasts of length $T \in [0.1, 0.5]$.

## 4   Conclusion

In this work, we have examined the efficacy of several commonly used numerical integration schemes for systems of SDEs
when applied to a standard benchmark toy model. This toy model, which we denote L96-s, has been contextualized in this study
as an ideal representation of a multiscale geophysical model; this represents a system in which the scale separation between
the evolution of fast and slow variables is taken to its asymptotic limit. This toy model, which is commonly used in benchmark
studies, represents a perfect-random model configuration for twin experiments. In this context, we have examined specifically:
(i) the modes and respective rates of convergence for each discretization scheme; and (ii) the biases introduced into ensemble-
based forecasting and DA due to discretization errors. In order to examine the efficacy of higher order integration methods, we
have furthermore provided a novel derivation of the strong order 2.0 Taylor scheme for systems with additive noise.

In the L96-s system, our numerical results have corroborated both the studies of Hatfield et al. (2018) and Frank and Gottwald
(2018). We find that the Euler-Maruyama scheme actually introduces a systematic bias in the ensemble forecasting in the L96-s
system. However, the effect of this bias on the DA cycle also strongly depends on the observation, and to a larger extent, model
uncertainty. When the intensity of the model uncertainty, governed by the strength of the diffusion coefficient, is increased,
we often see low-precision numerics performing comparably to higher-precision discretizations in the RMSE of filter twin
experiments. However, in lower model uncertainty regimes and with low precision numerics, the bias of the Euler-Maruyama
scheme is sufficient to produce filter divergence.

Weighing out the overall numerical complexity of each of the methods, and their respective accuracies in terms of mode
of convergence, it appears that a statistically robust configuration for twin experiments can be achieved by mixing integration
methods targeted for strong or weak convergence respectively. Specifically, the order 2.0 strong Taylor scheme provides good
performance in terms of strong convergence when the time-step is taken $\Delta \in \left[10^{-3}, 5 \times 10^{-3}\right]$. This guarantees a bound on the
path-discretization error below $10^{-3}$. On the other hand, the extremely generous coefficient in the bound for weak discretization
error for the Runge-Kutta scheme makes this method attractive for ensemble-based forecasting and deriving sample-based
statistics. While the performance depends strongly on the overall level of diffusion, a time-step of $\Delta \in \left[10^{-3}, 10^{-2}\right]$ bounds
the weak convergence discretization error by $10^{-3}$ for all of the studied diffusion levels.

Generally, it appears preferable to generate the ensemble forecast with the Runge-Kutta scheme and step size $\Delta = 10^{-3}$.
However, we find that the slight increase of error in the ensemble forecast by increasing the step size of the stochastic Runge-
Kutta scheme to $\Delta = 10^{-2}$ does not add any systematic bias. This is observed in terms of the short-time-scale forecast statistics,
the long-time-scale climatological statistics and in the filtering benchmarks. In all cases, it appears that additional variability
is introduced in the form of noise, yet this appears to be largely unbiased, random numerical error. In contrast, with the Euler-
Maruyama scheme we observe structural bias in the low-diffusion regimes which is enough to cause filter divergence when the



step-size is $\Delta = 10^{-2}$. Especially interesting, this is in the presence of what appears to be an artificial inflation of the ensemble spread with respect to the benchmark system.

Varying the accuracy of the generation of truth-twin, the results are largely the same as in a configuration with a finer step size. Disentangling a direct effect of the discretization error of the truth-twin from the effect of, e.g., observation error and the

diffusion in the process is difficult. Nonetheless, it appears that higher discretization accuracy of the truth-twin places a more stringent benchmark for filters in systems with less overall noise, especially due to the diffusion in the state evolution. There appears to be some relaxing of the RMSE benchmark when diffusion is high and the accuracy of truth-twin is low — in these cases we see lower RMSE overall for the coarsely evolved filters than in the benchmark system.

Therefore, we suggest a consistent and numerically efficient framework for twin experiments in which one produces the

truth-twin with the order strong 2.0 Taylor scheme and a time-step of $\Delta = 5 \times 10^{-3}$ and the stochastic Runge-Kutta scheme with a time-step of $\Delta = 10^{-2}$. In all diffusion regimes, this guarantees that the discretization error is bounded by $10^{-3}$, and most importantly, does not bias the filter results versus the more accurate benchmark system presented in this work. As the greatest computational expense for a twin experiment in the L96-s system is the ensemble generation, even as the system dimension $n$ grows, this allows for statistically robust and computationally cheap filter benchmarks to be performed.

As possible future work, we have not addressed the efficacy of weak schemes that are not guaranteed to converge to any path whatsoever. Particularly, of interest to the DA community and geophysical communities in general may be the following question: can generating ensemble forecasts with weak schemes reduce the overall cost of the ensemble forecasting step by neglecting the accuracy of any individual forecast, while maintaining a better accuracy and consistency of the ensemble-based statistics themselves. Weak schemes often offer many reductions in the numerical complexity due to reducing the goal to

producing an accurate forecast in distribution alone. Some methods that will be of interest for future study include, e.g., the order 3.0 weak Taylor scheme with additive noise (Kloeden and Platen, 2013)[see page 369] or the order 3.0 weak Runge-Kutta scheme page (Kloeden and Platen, 2013)[see page 488]. Additionally, it may be of interest to study other efficient, higher order strong Runge-Kutta schemes as discussed by Rößler (2010).

## 5 Code Availability

The current version of model is available from the project website: https://zenodo.org/badge/latestdoi/200740302 under the MIT License. The exact version of the model used to produce the results used in this paper is archived on Zenodo (Grudzien, 2019), as are scripts to run the model and produce the plots for all the simulations presented in this paper (Grudzien, 2019).

## Appendix A: Deriving the order 2.0 strong Taylor rule for L96-s

### A1 The abstract integration rule

We consider the SDE in Eq. (5), in the case where the noise covariance is scalar, though possibly of time dependent intensity: $\mathbf{S}$ will be assumed equal to $s(t)\mathbf{I}_n$ for some scalar function $s : \mathbb{R}^+ \rightarrow \mathbb{R}^+$. Suppose the state of the $i$-th component of the model





at time $t_k$ is given by $x^i(t_k) \triangleq x_k^i$ and $\Delta \triangleq t_{k+1} - t_k$. From page 359 of Kloeden and Platen (2013), the order 2.0 strong Taylor integration rule for $\boldsymbol{x}_k$ in Eq. (5) is written componentwise as

$$x_{k+1}^i = x_k^i + f^i \Delta + \frac{1}{2} \underline{\mathcal{L}}^0 f^i \Delta^2 \tag{A1a}$$

$$+ s W_\Delta^i + \sum_{j=1}^n \underline{\mathcal{L}}^j f^i J_{(j,0)} + \sum_{l,j=1}^n \underline{\mathcal{L}}^l \underline{\mathcal{L}}^j f^i J_{(l,j,0)} \tag{A1b}$$

where the right-hand side of Eq. (A1) is evaluated at $(\boldsymbol{x}_k, t_k)$ and the terms are defined as follows:

1. term (A1a) is the deterministic second order Taylor method. The differential operator $\underline{\mathcal{L}}^0$, defined on page 339 of Kloeden and Platen (2013), in the case of autonomous dynamics ($\boldsymbol{f}(\boldsymbol{x},t) = \boldsymbol{f}(\boldsymbol{x})$) with additive noise ($\mathbf{S}(\boldsymbol{x},t) = \mathbf{S}(t)$), reduces to

$$\underline{\mathcal{L}}^0 \triangleq \sum_{l=1}^n f^l \partial_{x^l} \tag{A2}$$

such that the term $\underline{\mathcal{L}}^0 f^i \equiv \boldsymbol{f}^{\mathrm{T}} \cdot \nabla f^i$;

2. $W_\Delta^i \triangleq W^i(\Delta) - W^i(0)$, where $W^i(t)$ is a one dimensional Wiener process. By definition, $W^i(0) = 0$ with probability one and $W_\Delta^i$ is a mean zero, Gaussian distributed random variable with variance equal to $\Delta$;

3. for each $l, j$, the differential operators $\underline{\mathcal{L}}^l, \underline{\mathcal{L}}^j$ are defined on page 339 of Kloeden and Platen (2013). In the case of additive noise with scalar covariance ($\mathbf{S}(\boldsymbol{x},t) = s(t)\mathbf{I}_n$), these operators reduce to

$$\underline{\mathcal{L}}^m \triangleq s \partial_{x^m} \quad \text{for any } 1 \le m \le n; \tag{A3}$$

4. for each $l, j$ the terms $J_{(j,0)}, J_{(l,j,0)}$, defined in pages 200 - 201 of Kloeden and Platen (2013), describe a recursive formulation of multiple Stratonovich integrals of the component random variables of $\boldsymbol{W}(t)$, over an interval of $[0, \Delta]$. These are given as

$$J_{(j,0)} \triangleq \frac{\Delta}{2}\left(W_\Delta^j + a_j\right); \tag{A4a}$$

$$J_{(l,j,0)} \triangleq \frac{\Delta}{2} W_\Delta^l W_\Delta^j - \frac{\Delta}{2}\left(a_j W_\Delta^l - a_l W_\Delta^j\right) + \Delta^2 A_{l,j}$$
$$- J_{(0,l,j)} - J_{(l,0,j)}; \tag{A4b}$$

$$J_{(0,l,j)} \triangleq \frac{\Delta}{3!} W_\Delta^l W_\Delta^j - \frac{\Delta}{\pi} W_\Delta^j b_l + \Delta^2 B_{l,j} - \frac{\Delta}{4} a_j W_\Delta^l$$
$$+ \frac{\Delta}{2\pi} W_\Delta^l b_j + \Delta^2 C_{l,j} + \frac{\Delta^2}{2} A_{l,j}; \tag{A4c}$$

$$J_{(l,0,j)} \triangleq \frac{\Delta}{3!} W_\Delta^l W_\Delta^j + \frac{a_l}{2} J_{(0,j)} + \frac{\Delta}{2\pi} W_\Delta^j b_l - \Delta^2 B_{l,j}$$
$$- \frac{\Delta}{4} a_j W_\Delta^l + \frac{\Delta}{2\pi} W_\Delta^l b_j; \tag{A4d}$$

$$J_{(0,j)} \triangleq \frac{\Delta}{2}\left(W_\Delta^j - a_j\right); \tag{A4e}$$





5. coefficients $A_{l,j}, B_{l,j}, C_{l,j}, a_l, a_j, b_l, b_j$, in Eqs. (A4a) - (A4e) are defined on pages 198-199 and 201, via the component-
wise Karhunen-Loève Fourier expansion (Kloeden and Platen, 2013)[see pages 70-71] of a Brownian bridge process
$\boldsymbol{B}(t) \triangleq \boldsymbol{W}_t - \frac{t}{\Delta}\boldsymbol{W}_\Delta$, for $0 \le t \le \Delta$. We write the expansion of $\boldsymbol{B}(t)$ component-wise as

$$B^j(t) = \frac{a_j}{2} + \sum_{r=1}^{\infty}\left[a_{j,r}\cos\left(\frac{2r\pi t}{\Delta}\right) + b_{j,r}\sin\left(\frac{2r\pi\tau}{\Delta}\right)\right]. \tag{A5}$$

The random Fourier coefficients $a_{j,r}, b_{j,r}$ are defined for each $r \in \mathbb{N}^+$,

$$a_{j,r} \triangleq \frac{2}{\Delta}\int_0^\Delta \left(W_\tau^j - \frac{\tau}{\Delta}W_\Delta^j\right)\cos\left(\frac{2r\pi\tau}{\Delta}\right)\mathrm{d}\tau, \tag{A6}$$

$$b_{j,r} \triangleq \frac{2}{\Delta}\int_0^\Delta \left(W_\tau^j - \frac{\tau}{\Delta}W_\Delta^j\right)\sin\left(\frac{2r\pi\tau}{\Delta}\right)\mathrm{d}\tau, \tag{A7}$$

as pairwise independent, Gaussian distributed random variables,

$$a_{j,r}, b_{j,r} \sim N\left(0, \frac{\Delta}{2\pi^2 r^2}\right). \tag{A8}$$

The convergence of the right-hand side of Eq. (A5) to the left-hand side is in the mean-square sense ($L^2$ norm) and
uniform in $t$. From the Fourier coefficients, for each $j$, we define $a_j, b_j$,

$$a_j \triangleq \frac{2}{\Delta}\int_0^\Delta \left(W_\tau^j - \frac{\tau}{\Delta}W_\Delta^j\right)\mathrm{d}\tau, \qquad\qquad b_j \triangleq \sum_{r=1}^\infty \frac{1}{r}b_{j,r}, \tag{A9}$$

and the auxiliary coefficients as,

$$A_{l,j} \triangleq \frac{\pi}{\Delta}\sum_{r=1}^\infty r\left(a_{l,r}b_{j,r} - a_{j,r}b_{l,r}\right), \tag{A10}$$

$$B_{l,j} \triangleq \frac{1}{2\Delta}\sum_{r=1}^\infty \left(a_{l,r}a_{j,r} + b_{l,r}b_{j,r}\right), \tag{A11}$$

$$C_{l,j} \triangleq \frac{-1}{\Delta}\sum_{r,q=1 \,\&\, r\neq q}^\infty \frac{r}{r^2 - q^2}\left(ra_{l,r}a_{j,q} + qb_{l,r}b_{j,q}\right). \tag{A12}$$

Expanding Eq. (A1) in the above defined terms gives an explicit integration rule that has strong convergence on order 2.0 in
the maximum step size. The subject of the next section is utilizing the symmetry and the constant/ vanishing derivatives of the
Lorenz-96 model to derive significant reductions to the above general rule.





## A2 Deriving reductions to the rule for L96-s

We note that

$$
\frac{\partial f^i}{\partial x^j} = \begin{cases} -x^{i-1} & j = i-2 \\ x^{i+1} - x^{i-2} & j = i-1 \\ -1 & j = i \\ x^{i-1} & j = i+1 \\ 0 & \text{else,} \end{cases} \tag{A13}
$$

from which we derive

$$
\frac{\partial^2 f^i}{\partial x^l \partial x^j} = \begin{cases} -1 & j = i-2,\ l = i-1 \\ -1 & j = i-1,\ l = i-2 \\ 1 & j = i-1,\ l = i+1 \\ 1 & j = i+1,\ l = i-1 \\ 0 & \text{else.} \end{cases} \tag{A14}
$$

The constancy of the second derivatives in Eq. (A14) will allow us to simplify the expressions in term (A1b). Specifically,
notice that

$$
\underline{\mathcal{L}}^l \underline{\mathcal{L}}^j f^i J_{(l,j,0)} = s^2 \frac{\partial^2 f^i}{\partial x^l \partial x^j} J_{(l,j,0)}, \tag{A15}
$$

such that the sum $\sum_{l,j=1}^n \underline{\mathcal{L}}^l \underline{\mathcal{L}}^j f^i J_{(l,j,0)}$ reduces to

$$
\begin{aligned}
\sum_{l,j=1}^n & \underline{\mathcal{L}}^l \underline{\mathcal{L}}^j f^i J_{(l,j,0)} \\
=& s^2 \left( J_{(i-1,i+1,0)} + J_{(i+1,i-1,0)} \right) \\
& - s^2 \left( J_{(i-2,i-1,0)} + J_{(i-1,i-2,0)} \right).
\end{aligned} \tag{A16}
$$

We are thus interested in reducing the terms of Eq. (A16) via anti-symmetry within $J(l,j,0)$ with respect to the arguments $l,j$.
We note that

$$
A_{l,j} = -A_{j,l}, \tag{A17}
$$

$$
(a_j W_\Delta^l - a_l W_\Delta^j) = -(a_l W_\Delta^j - a_j W_\Delta^l), \tag{A18}
$$

and combining these relationships with the definition in Eq. (A4b), we find

$\quad J_{(l,j,0)} + J_{(j,l,0)} = \Delta W_\Delta^l W_\Delta^j - \left[ \left( J_{(l,0,j)} + J_{(0,l,j)} \right) + \left( J_{(j,0,l)} + J_{(0,j,l)} \right) \right].$ $\qquad$ (A19)





Notice that from Eqs. (A4c) and (A4d), the sum $J_{(l,0,j)} + J_{(0,l,j)}$ contains the terms on the left-hand side of Eq. (A20),

$$\frac{\Delta}{3!} W_\Delta^j W_\Delta^l + \frac{\Delta}{3!} W_\Delta^j W_\Delta^l = \frac{\Delta}{3} W_\Delta^j W_\Delta^l; \tag{A20a}$$

$$\Delta^2 B_{l,j} - \Delta^2 B_{l,j} = 0; \tag{A20b}$$

$$-\frac{\Delta}{4} a_j W_\Delta^l - \frac{\Delta}{4} a_j W_\Delta^l = -\frac{\Delta}{2} a_j W_\Delta^l; \tag{A20c}$$

$$\frac{\Delta}{2\pi} W_\Delta^l b_j + \frac{\Delta}{2\pi} W_\Delta^l b_j + \frac{\Delta}{2\pi} W_\Delta^j b_l - \frac{\Delta}{\pi} W_\Delta^j b_l = \frac{\Delta}{\pi} W_\Delta^l b_j - \frac{\Delta}{2\pi} W_\Delta^j b_l. \tag{A20d}$$


Combining terms as in the left-hand side of Eq. (A20), and substituting the right-hand side of the terms in Eq. (A20) we derive that

$$J_{(l,0,j)} + J_{(0,l,j)} = \frac{\Delta}{3} W_\Delta^l W_\Delta^j \tag{A21a}$$

$$+ \frac{1}{2} a_l J_{(0,j)} - \frac{\Delta}{2} a_j W_\Delta^l \tag{A21b}$$

$$+ \frac{\Delta}{\pi} \left( W_\Delta^l b_j - \frac{1}{2} W_\Delta^j b_l \right) \tag{A21c}$$


$$+ \Delta^2 \left( C_{l,j} + \frac{1}{2} A_{l,j} \right). \tag{A21d}$$

Note then, from Eq. (A19), we need to combine the terms of the symmetric sum in $l, j$,

$$\left( J_{(l,0,j)} + J_{(0,l,j)} \right) + \left( J_{(j,0,l)} + J_{(0,j,l)} \right). \tag{A22}$$

We thus use the anti-symmetry in the terms in Eq. (A21) to make further reductions. Note that from Eq. (A4e) we have


$$\frac{1}{2} a_l J_{(0,j)} = \frac{\Delta}{4} \left( a_l W_\Delta^j - a_l a_j \right). \tag{A23}$$

Therefore, the symmetric sum in $l, j$ of $\frac{1}{2} a_l J_{(0,j)}$ is given by

$$\frac{1}{2} a_l J_{(0,j)} + \frac{1}{2} a_j J_{(0,l)} = \frac{\Delta}{4} \left( a_l W_\Delta^j - a_l a_j + a_j W_\Delta^l - a_j a_l \right)$$

$$= \frac{\Delta}{4} \left( a_l W_\Delta^j + a_j W_\Delta^l \right) - \frac{\Delta}{2} a_l a_j. \tag{A24}$$

Thus using Eq. (A24), the symmetric sum of term (A21b) in $l, j$ equals


$$\frac{\Delta}{4} \left( a_l W_\Delta^j + a_j W_\Delta^l \right) - \frac{\Delta}{2} a_l a_j - \frac{\Delta}{2} \left( a_j W_\Delta^l + a_l W_\Delta^j \right) = \frac{-\Delta}{4} \left( a_l W_\Delta^j + a_j W_\Delta^l \right) - \frac{\Delta}{2} a_l a_j. \tag{A25}$$

Likewise, taking the sum of term (A21c) symmetrically in $l, j$ equals

$$\frac{\Delta}{\pi} \left( W_\Delta^l b_j - \frac{1}{2} W_\Delta^j b_l + W_\Delta^j b_l - \frac{1}{2} W_\Delta^l b_j \right)$$

$$= \frac{\Delta}{2\pi} \left( W_\Delta^l b_j + W_\Delta^j b_l \right). \tag{A26}$$





Recalling the anti-symmetry of $A_{l,j}$ and the substitutions in Eqs. (A25) and (A26), we combine the terms $J_{(l,0,j)} + J_{(0,l,j)} + J_{(j,0,l)} + J_{(0,j,l)}$ to derive

$$
\begin{aligned}
&J_{(l,0,j)} + J_{(0,l,j)} + J_{(j,0,l)} + J_{(0,j,l)} \\
=& \frac{2\Delta}{3} W_\Delta^l W_\Delta^j - \frac{\Delta}{4} \left( W_\Delta^l a_j + W_\Delta^j a_l \right) \\
&- \frac{\Delta}{2} a_l a_j + \frac{\Delta}{2\pi} \left( W_\Delta^l b_j + W_\Delta^j b_l \right) \\
&+ \Delta^2 \left( C_{l,j} + C_{j,l} \right).
\end{aligned} \tag{A27}
$$

Finally, using Eq. (A27), let us define the symmetric function in $(l,j)$,

$$
\begin{aligned}
\Psi_{(l,j)} =& \Delta W_\Delta^l W_\Delta^j \\
&- \left[ \left( J_{(l,0,j)} + J_{(0,l,j)} \right) + \left( J_{(j,0,l)} + J_{(0,j,l)} \right) \right] \\
=& \frac{\Delta}{3} W_\Delta^l W_\Delta^j + \frac{\Delta}{4} \left( W_\Delta^l a_j + W_\Delta^j a_l \right) + \frac{\Delta}{2} a_l a_j \\
&- \frac{\Delta}{2\pi} \left( W_\Delta^l b_j + W_\Delta^j b_l \right) - \Delta^2 \left( C_{l,j} + C_{j,l} \right);
\end{aligned} \tag{A28}
$$

from the above definition and Eq. (A16), we recover the expression

$$
\sum_{l,j=1}^{n} \mathcal{L}^l \mathcal{L}^j f^i J_{(l,j,0)} = s^2 \left[ \Psi_{(i-1,i+1)} - \Psi_{(i-2,i-1)} \right]. \tag{A29}
$$

Furthermore, define the random vectors

$$
\boldsymbol{\Psi}_+ \triangleq \begin{pmatrix} \Psi_{(n,2)} \\ \vdots \\ \Psi_{(n-1,1)} \end{pmatrix}, \qquad \boldsymbol{\Psi}_- \triangleq \begin{pmatrix} \Psi_{(n-1,n)} \\ \vdots \\ \Psi_{(n-2,n-1)} \end{pmatrix}, \tag{A30a}
$$

$$
\boldsymbol{J}_\Delta \triangleq \begin{pmatrix} \frac{\Delta}{2} \left( W_\Delta^1 + a_1 \right) \\ \vdots \\ \frac{\Delta}{2} \left( W_\Delta^n + a_n \right) \end{pmatrix}. \tag{A30b}
$$

Using the above definitions, we can write the integration rule in a matrix form as

$$
\boldsymbol{x}_{k+1} = \boldsymbol{x}_k + \boldsymbol{f}\Delta + \frac{\Delta^2}{2} \nabla \boldsymbol{f} \cdot \boldsymbol{f} \tag{A31a}
$$

$$
+ s\boldsymbol{W}_\Delta + s\nabla \boldsymbol{f} \cdot \boldsymbol{J}_\Delta + s^2 \left( \boldsymbol{\Psi}_+ - \boldsymbol{\Psi}_- \right). \tag{A31b}
$$

Once again, the term (A31a) is the standard deterministic order 2.0 Taylor rule but written in matrix form. On the other hand, the additional term (A31b) resolves at second order the SDE form of L96 with additive noise of scalar covariance (L96-s).





## A3 Finite approximation and numerical computation

So far we have only presented an abstract integration rule that implicitly depends on infinite series of random variables. Truncating the Fourier series for the components of the Brownian bridge in Eq. (A5), we define a random process


$$W_t^{j,p} \triangleq \frac{t}{\Delta} W_\Delta^j + \frac{a_j}{2}$$
$$+ \sum_{r=1}^{p} \left[ a_{j,r} \cos\left(\frac{2r\pi t}{\Delta}\right) + b_{j,r} \sin\left(\frac{2r\pi \tau}{\Delta}\right) \right], \tag{A32}$$

from which we will define a numerical integration rule, depending on the order of truncation $p$. Key to the computation of the rule is that, by way of the approximations in pages 202 - 204 of Kloeden and Platen (2013), it is representable as a function of mutually independent, standard Gaussian random variables. We will denote these standard Gaussian random variables as

$\xi_j, \zeta_{j,r}, \eta_{j,r}, \mu_{j,p}$ and $\phi_{j,p}$, and for each $j = 1, \cdots n$, $r = 1, \cdots, p$ and all $p \in \mathbb{N}^+$, we define

$$\xi_j \triangleq \frac{1}{\sqrt{\Delta}} W_\Delta^j, \qquad\qquad \mu_{j,p} \triangleq \frac{1}{\sqrt{\Delta \rho_p}} \sum_{r=p+1}^{\infty} a_{j,r}, \tag{A33a}$$

$$\zeta_{j,r} \triangleq \sqrt{\frac{2}{\Delta}} \pi r a_{r,j}, \qquad\qquad \phi_{j,p} \triangleq \frac{1}{\sqrt{\Delta \alpha_p}} \sum_{r=p+1}^{\infty} \frac{1}{r} b_{j,r}, \tag{A33b}$$

$$\eta_{j,r} \triangleq \sqrt{\frac{2}{\Delta}} \pi r b_{r,j}, \tag{A33c}$$

where

$$\rho_p \triangleq \frac{1}{12} - \frac{1}{2\pi^2} \sum_{r=1}^{p} \frac{1}{r^2}, \qquad\qquad \alpha_p \triangleq \frac{\pi^2}{180} - \frac{1}{2\pi^2} \sum_{r=1}^{p} \frac{1}{r^4}. \tag{A34}$$

It is important to note that while $\mu_{j,p}, \phi_{j,p}$ are defined as an infinite linear combination of the random Fourier coefficients, we take $\mu_{j,p}$ and $\phi_{j,p}$ as drawn iid from the standard Gaussian distribution and use their functional relationship to the Fourier coefficients to approximate the Stratonovich integral. The coefficients $\rho_p$ and $\alpha_p$ normalize the variance of the remainder term in the truncation of the Brownian bridge process to the finite sum $W_t^{j,p}$. Using the above defined random variables in Eq.

(A33), and auxiliary, deterministic variables in Eq. (A34), we will define the $p$-th approximation of the multiple Stratonovich integrals in Eqs. (A4e) - (A4c).

For any $p \in \mathbb{N}^+$, and for each $j = 1, \cdots n$, we can recover the term $b_j$ directly from the functional relationships in Eq. (A33), and $a_j$ by the relationship on page 203 of Kloeden and Platen (2013),

$$a_j \triangleq -2\sqrt{\Delta \rho_p} \mu_{j,p} - \frac{\sqrt{2\Delta}}{\pi} \sum_{r=1}^{p} \frac{\zeta_{j,r}}{r}; \tag{A35a}$$

$$b_j \triangleq \sqrt{\Delta \alpha_p} \phi_{j,p} + \sqrt{\frac{\Delta}{2\pi^2}} \sum_{r=1}^{p} \frac{1}{r^2} \eta_{j,r}. \tag{A35b}$$

The auxiliary function $C_{l,j}$ is truncated at the $p$-th order, defined on page 203 of Kloeden and Platen (2013), as

$$C_{l,j}^p \triangleq \frac{-1}{2\pi^2} \sum_{r,q=1 \,\&\, r \neq q}^{p} \frac{r}{r^2 - q^2} \left( \frac{1}{q} \zeta_{l,r} \zeta_{j,q} + \frac{1}{r} \eta_{l,r} \eta_{j,q} \right). \tag{A36}$$





While the choice of $p$ modulates the order of approximation of the Stratonovich integrals, it is important to note, in our case the choice of $p > 1$ is unnecessary. Actually, all terms in the Stratonovich integrals in the integration rule we have derived are exact except for the terms of $C_{l,j}^p$. Up to a particular realization of the random variables, $a_j$ and $b_j$ are constructed identically from the full Fourier series. It is thus only the terms of $C_{i,j}^p$ that are truncated, and this approximation appears at order 2.0 in the integration step. Therefore, up to a constant that depends on $p$, the approximation error of the Stratonovich integrals is also at order 2.0. Note that, by definition, when $p = 1$ $C_{l,j}^p \equiv 0$ for all $l, j = 1, \cdots, n$. Therefore, we may eliminate this term in our finite approximation without loss of the order of convergence.

For simplicity, at each integration step for each $j = 1, \cdots n$, $r = p = 1$, we may draw $n \times (2p + 3) = n \times 5$ iid standard Gaussian random variables, $\xi_j, \zeta_{j,r}, \eta_{j,r}, \mu_{j,p}$ and $\phi_{j,p}$, to obtain an approximation of the recursive Stratonovich integrals in Eq. (A4). For each $j$ we will make substitutions as described in Eq. (A34) - (A35) to obtain the final integration rule. Using the simplifications made to the rule in section A.A2, and the above discussion, we define

$$\Psi_{(l,j)}^p \triangleq \frac{\Delta^2}{3} \xi_l \xi_j + \frac{\Delta^{\frac{3}{2}}}{4} \left( \xi_l a_j + \xi_j a_l \right) + \frac{\Delta}{2} a_l a_j$$

$$- \frac{\Delta^{\frac{3}{2}}}{2\pi} \left( \xi_l b_j + \xi_j b_l \right). \tag{A37}$$

Finally, we define the following random vectors

$$\boldsymbol{\Psi}_+^p \triangleq \begin{pmatrix} \Psi_{(n,2)}^p \\ \vdots \\ \Psi_{(n-1,1)}^p \end{pmatrix}, \qquad\qquad \boldsymbol{\Psi}_-^p \triangleq \begin{pmatrix} \Psi_{(n-1,n)}^p \\ \vdots \\ \Psi_{(n-2,n-1)}^p \end{pmatrix}, \tag{A38a}$$

$$\boldsymbol{J}_\Delta^p \triangleq \begin{pmatrix} \frac{\Delta}{2} \left( \sqrt{\Delta} \xi_1 + a_1 \right) \\ \vdots \\ \frac{\Delta}{2} \left( \sqrt{\Delta} \xi_n + a_n \right) \end{pmatrix}, \qquad\qquad \boldsymbol{\xi} = \begin{pmatrix} \xi_1 \\ \vdots \\ \xi_n \end{pmatrix} \tag{A38b}$$

such that we obtain the integration rule in matrix form,

$$\boldsymbol{x}_{k+1} = \boldsymbol{x}_k + \boldsymbol{f} \Delta + \frac{\Delta^2}{2} \nabla \boldsymbol{f} \cdot \boldsymbol{f} \tag{A39a}$$

$$+ s\sqrt{\Delta} \boldsymbol{\xi} + s \nabla \boldsymbol{f} \cdot \boldsymbol{J}_\Delta^p + s^2 \left( \boldsymbol{\Psi}_+^p - \boldsymbol{\Psi}_-^p \right). \tag{A39b}$$

The constants $\rho_p$ and $\alpha_p$ can be computed once for all steps with truncation can be taken at $p = 1$ such that

$$\rho_p = \frac{1}{12} - \frac{1}{2\pi^2}; \qquad\qquad \alpha_p = \frac{\pi^2}{180} - \frac{1}{2\pi^2}. \tag{A40}$$

Then, for each step of size $\Delta$, we can follow the rule outlined in Eq. (14).

*Author contributions.* CG derived the order 2.0 Taylor discretization for the L96-s model, developed all model code and processed all data. CG and MB reviewed and refined mathematical results together. All authors contributed to the design of numerical experiments. CG wrote the manuscript with contributions from MB and AC



*Competing interests.* The authors declare that they have no conflict of interest.

*Acknowledgements.* This work benefited from funding by the project REDDA of the Norwegian Research Council under contract 250711.
800 This work benefited significantly from CEREA hosting Colin Grudzien as a visiting researcher in 2018, during his postdoctoral appointment
at NERSC. CEREA is a member of the Institut Pierre-Simon Laplace (IPSL). The authors would like to thank Peter Kloeden, Eckhard Platen
and Paul Hurtado for their correspondence and suggestions on this work.



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
