# Peer review of "On the numerical integration of the Lorenz-96 model, with scalar additive noise, for benchmark twin experiments"

_Geoscientific Model Development, 2019_

## Referee Comment (RC1) · Anonymous Referee #1 · 3 Jan 2020

The authors make a thorough investigation about the effects of numerical discretization error that practitioners often ignore in the data assimilation (DA) community. They study the bias, caused by the time-discretization error, on the ensemble-based statistics and filtering performance for twin experiments in the Lorenz-96 system with an additive noise. Three numerical schemes are investigated: the Euler-Maruyama scheme, a Taylor scheme of strong order 2, and a Runge-Kutta scheme, which are of strong order 1.0, 2.0 and 1.0 (also, weak order 1.0, 2.0 and 1.0) respectively. The authors tested forecast ensemble statistics, and ensemble data assimilation with different levels of observational noise, with different levels of diffusion for all the tests.

[Figure]

The paper is well-written and pleasant to read. The results provide a guidance on designing twin experiments in data assimilation. I recommend its publication. The following are a few points that the authors may want to address before publication.

Suggestions:

- In Section 3.1, there should be an explanation on that, in the order tests with different time-steps, how the Taylor scheme in Section 2.5 is implemented. In particular, how the random vectors are added up to consistently approximate the multiple stochastic integral (which leads to the Taylor scheme).

- In the convergence order tests, three time-step sizes are used, and this is not very convincing, in particular in Figure 2. It would be more convincing to use about four or five time-step sizes. Instead of $\Delta = \{10^{-i}\}_{i=1,2,3}$, the authors may use other bases, e.g. $\Delta = \{2^{-i}\}_{i=5,\ldots,9}$

- The conclusion seems weak and too specific (in particular, the time-steps in this specific test setting are presented without further analysis). It might be helpful to explore more on the balance between the level of diffusion and the right-hand-side of the equation, as well as the Lyapunov exponent of the system.

- there might be typos in the indexing of $i, j, b$ in Eq.(15-16).

---

## Referee Comment (RC2) · Anonymous Referee #2 · 9 Jan 2020

This paper introduces the L96-s system as a stochastic differential equation with drift coefficient given by the Lorenz-96 equations and uncorrelated Brownian noise process multiplied a possibly time varying scalar diffusion coefficient. This is a timely contribution as many existing papers, especially in the data assimilation literature, have been making various ad hoc stochastic modification to the Lorenz-96 equations without introducing a rigorous SDE. The author's consider several consistent methods of numerically simulating the L96-s, including the easy to implement (but rarely used) Euler-Maruyama, and the even more rarely used Milstein scheme. They then consider the commonly used Runge-Kutta scheme, and illustrate how to correctly implement the stochastic forcing in order to have strong order-1.0 convergence (the various schemes

in the literature can be very different so this in itself is a valuable contribution). The authors also introduce a strong order 2.0 method based on a Taylor scheme. A thorough comparison is made between these methods, including how ensemble spread differs, which is a useful intuition for practitioners. One thing that would have been interesting to see (although perhaps difficult to produce) would be to compare trajectories generated with the various schemes but with the same noise realization (probably requiring Brownian bridges due to the different interior point samples).

The second part of the paper studies the effect of different numerical schemes on data assimilation. Using their Taylor 2.0 scheme with a fine discretization to generate the truth, they then attempt assimilation with various other schemes. The analysis is very thorough, and includes a range of diffusion levels and observation noise levels. It would be interesting to see if introducing inflation into the data assimilation scheme (artificially increasing the diffusion coefficient used by the filter to compensate for model error) could have compensated for the large errors introduced by the Euler-Maruyama scheme at the coarse time scale.

---

## Author Comment (AC1) · 8 Feb 2020

**1   Introduction**

The authors would like to thank the referees for their time and their valuable comments – their reviews have been very useful for clarifying several points in the text, and for improving the robustness of our results. We will highlight the main change to the text in the following, and address smaller points in the section thereafter.

[Figure]

**2 Main change**

Our largest change to the manuscript is to address the following comments:

- **Referee 1:**

    In Section 3.1, there should be an explanation on that, in the order tests with different time-steps, how the Taylor scheme in Section 2.5 is implemented. In particular, how the random vectors are added up to consistently approximate the multiple stochastic integral (which leads to the Taylor scheme).

- **Referee 2:**

    One thing that would have been interesting to see (although perhaps difficult to produce) would be to compare trajectories generated with the various schemes but with the same noise realization (probably requiring Brownian bridges due to the different interior point samples).

In our original manuscript, we included a description on how we used the Brownian motion realizations, discretized on the fine scale for the reference path, to consistently define the coarse discretization methods and the Brownian bridge for the Taylor scheme. However, this earlier exposition was not written in a clear and obvious way, and to account for this issue we have expanded and refined this description in our new Appendix B.

In addition, we have clarified the exposition of the overall experimental setup in Section 3.1 to make this more understandable. This is in conjunction with revising this experimental setup as in the following suggestion:

- **Referee 1:**

**[GMDD](**

Interactive
comment
> In the convergence order tests, three time-step sizes are used, and this is not very convincing, in particular in Figure 2. It would be more convincing to use about four or five time-step sizes.

We appreciate this suggestion, and we believe our new Figs. 1 - 2 are indeed a nice improvement over the previous versions. We have chosen to use the discretization time steps $\{2^{-q}\}_{q=5}^{9}$ exactly as was suggested. In order to do so, we have also changed the step size of the finely-discretized reference path to $2^{-23}$ and the time horizon $T = 0.125$. These changes are performed in order to keep the discretization of the Brownian motion realizations consistent between the fine and coarse step-sizes, as well as to consistently define the Brownian Bridges in between the coarse steps. The methods for doing so are detailed in our revised Section 3.1 and our new Appendix B.

Updates are performed to Table 1, for the estimated coefficients for the bounds on the expected discretization errors based on these new experiments. Statements that used calculations based on Table 1 have been updated. Specifically, we have found that the estimated discretization error for the Taylor scheme, using a step size $5 \times 10^{-3}$, is bounded by approximately $0.001075$ across all difussion regimes. For this reason, we rephrase earlier conclusions stating that the error was bounded by $10^{-3}$ to state that the error is close to $10^{-3}$ across all regimes.

The updated figures are included in this comment at the bottom of the text.

**3   Minor changes**

- **Referee 1:**

    There might be typos in the indexing of $i, j, b$ in Eq.(15-16).

    **Response**

We believe that the indices are correct, but we note that we suppressed the use of the index $b$ in the terms on the right-hand-side. To make this more clear, we have included text before to indicate that we have suppressed indices, and after the equations to clarify which ones are suppressed.

- **Referee 1:**

  The conclusion seems weak and too specific (in particular, the time-steps in this specific test setting are presented without further analysis). It might be helpful to explore more on the balance between the level of diffusion and the right-hand-side of the equation, as well as the Lyapunov exponent of the system.

  **Response**

  We appreciate this suggestion and to address this we have slightly expaned the conclusion to discuss when the state evolution is drift dominated versus diffusion dominated. However, relating this specifically to the Lyapunov time of the system appears to us to be a subtle question and one that we are not prepared to answer at this stage of the work. We believe that this is a worthwhile question for future investigation, but one that goes beyond our current scope.

- **Referee 2:**

  It would be interesting to see if introducing inflation into the data assimilation scheme (artificially increasing the diffusion coefficient used by the filter to compensate for model error) could have compensated for the large errors introduced by the Euler-Maruyama scheme at the coarse time scale.

**Response**

We appreciate the suggestion, and we believe that this is an interesting question to investigate. However, at the moment we feel that this will go beyond the scope of the work.

[Figure]

[Figure]

**Fig. 1.** Strong Convergence Benchmark - Revised

[Figure]

**Fig. 2.** Weak Convergence Benchmark - Revised

---

## Author Response (AR2)

**1 Introduction**

We thank the editor for these comments and for improving the quality of the text. Below are our responses.

- **Editor:** I suggest adding a relevant citation near to the sentence starting on line 48.

  We have moved the reference Hatfield et al. from the end of the paragraph to the earlier line 48 to make more clear that this was the reference to which we were referring.

- **Editor:** "Fokker-Planck" is misspelled

  This has been fixed, thank you for catching this error.

- **Editor:** Is the phrasing "of the order" intended in definitions 1 and 2?

  We have chosen to adopt the phrasing "of the order" throughout the text in our revision rather than "on the order", this is written as intended.

- **Editor:** I suggest clarifying the sentences starting on lines 362 and 366 – against what is the RK scheme error compared to here?

  We thank the editor for noticing that this sentence was written poorly. This refers to the effect of the constant $C$ on the overall discretization error in the scheme. Whereas in the Taylor scheme this constant penalizes the method, raising the overall discretization error by an order of magnitude, the constant in the Runge-Kutta scheme actually reduces the overall discretization error by about an order of magnitude. This has been clarified in the text.

- **Editor:** The term "almost always" should be avoided on lines 430-431.

  This has been removed.

[revised manuscript text omitted]